# Public perceptions and support of climate intervention technologies across the Global North and Global South

Chad M. Baum [1] ✉, Livia Fritz [1], Sean Low[1] & Benjamin K. Sovacool[1,2,3]

Novel, potentially radical climate intervention technologies like carbon dioxide removal and solar geoengineering are attracting attention as the adverse impacts of climate change are increasingly felt. The ability of publics, particularly in the Global South, to participate in discussions about research, policy, and deployment is restricted amidst a lack of familiarity and engagement. Drawing on a large-scale, cross-country exercise of nationally representative surveys ($N = 30,284$) in 30 countries and 19 languages, this article establishes the first global baseline of public perceptions of climate-intervention technologies. Here, we show that Global South publics are significantly more favorable about potential benefits and express greater support for climate-intervention technologies. The younger age and level of climate urgency and vulnerability of these publics emerge as key explanatory variables, particularly for solar geoengineering. Conversely, Global South publics express greater concern that climate-intervention technologies could undermine climate-mitigation efforts, and that solar geoengineering could promote an unequal distribution of risks between poor and rich countries.

Recent years have provided growing evidence of the increasing impacts of climate change and the mounting costs of adaptation. Reflecting the historical inequalities of how and where emissions were generated, the impacts and human costs of climate change have to be unjustly borne by those in the Global South[1,2]. The WMO's Atlas[2] of climate-related mortality and economic losses highlights the growing damage of natural disasters, particularly in the Global South, and the expected proliferation of loss and damage related to extreme heat, drought, and a litany of other risks in the coming years. Set against the ever-greater concern around the insufficient pace of emissions reductions, stronger consideration is being given to more novel, often radical climate-intervention technologies[3–5]. The latter consists of carbon dioxide removal (CDR) approaches, more familiar ecosystem-based forms of afforestation and soil carbon sequestration as well as novel engineered approaches such as direct air capture, and solar radiation modification (SRM) methods such as stratospheric aerosol injection and space-based geoengineering.

As the available carbon budget dwindles and the prospect of emissions and temperature overshoots become more likely[6], climate-intervention technologies, in spite of their risks and prevailing uncertainty[7–9], are attracting further assessment – especially if they are to be scaled up and widely deployed. However, even as such techniques enter the radar of scientists and policymakers, societal debates involving the wider public are only slowly emerging. Thus, the public is substantially unfamiliar with these technologies and what they entail. Since the first public-perceptions study examining climate-intervention technologies was published twelve years ago[10], there have been several dozen publications, mostly of a quantitative nature[11]. In this time, there is repeated observation of the public's lack of familiarity and how little this has changed in the past decade[12–15].

[1]Department of Business Development and Technology, Aarhus University, Birk Centerpark 15, 7400 Herning, Denmark. [2]Science Policy Research Unit (SPRU), University of Sussex Business School, Jubilee Building, Arts Rd, Falmer, Brighton BN1 9SL, UK. [3]Department of Earth and Environment, Boston University, 685 Commonwealth Ave, Boston, MA 02215, USA. ✉e-mail: cmbaum@btech.au.dk

Substantive obstacles thus persist for engaging the global public in deliberation, appreciating their concerns and addressing any confusion, along with running the risk that discussions and decision-making take place without the public.

Coupled with the overall need for more knowledge about public perceptions, there are various gaps in the literature that require addressing. First, few studies have employed a cross-country design to compare and contrast perceptions[16–20]. Second, only a handful of surveys assess perceptions of more than one technology – even then this is done by assigning one technology to each participant. Third, and most problematically, there has been very little engagement with actors in the Global South. This represents a severe flaw given that researchers and decision-makers in such regions will likely have divergent perspectives, use their own criteria to evaluate such technologies – not to mention arguments it is the severe impacts confronting the Global South which impart greater urgency to solar-geoengineering activities[21–25]. Also, little of the literature explicitly centers public perceptions in the Global South[16–18,20], with just three countries (United States of America, United Kingdom, Germany) accounting for more than 60% of the times that a country features in the research[11]. If the research and understanding of climate-intervention technologies is to become more inclusive and benefit from deliberation by diverse actors, from disparate perspectives, there is an urgent need for research investigating public perceptions of climate-intervention technologies across multiple countries in the Global South which would facilitate a comparison with the Global North.

This study presents the findings of a large-scale, cross-country, set of nationally representative surveys ($n = 30,284$ participants, with at least 1000 in each country) of the public perceptions of CDR and SRM technologies in 30 countries and conducted in 19 languages (Fig. 1; see "Methods"). Survey samples were nationally representative of country populations in terms of age (between 18 and 74), gender, and geographic region and with broad quotas for education and income. By conducting surveys in a range of countries, the results provide a first baseline of climate-intervention perceptions at a global level. We examine key areas of agreement and disagreement between publics across the Global North and Global South, in terms of relative preferences for certain types of technologies (e.g., for ecosystem-based CDR, and CDR over SRM) and policy approaches that would facilitate the research and development of, rather than prohibit, climate-intervention technologies. We highlight the differences between these publics given the emergent debates, and lack of research, on public perceptions of climate-intervention technologies in the Global South, along with the prominence of this distinction in the literature.

The survey instrument examined ten climate-intervention technologies that have been the focus of most research and discussion, categorized into three technology categories, with participants randomly assigned to one of the following groups: SRM (stratospheric aerosol injection, space-based geoengineering, marine cloud brightening); ecosystem-based CDR (afforestation and reforestation, soil carbon sequestration, marine biomass and blue carbon); and engineered CDR (direct air capture with carbon storage (DACCS), bioenergy with carbon capture and storage (BECCS), enhanced weathering, biochar). The distinction between ecosystem-based CDR and engineered CDR might be imperfect, but we defend it given that the categories entail different kinds of resource and energy demands as well as how space is used, such that these differences may be

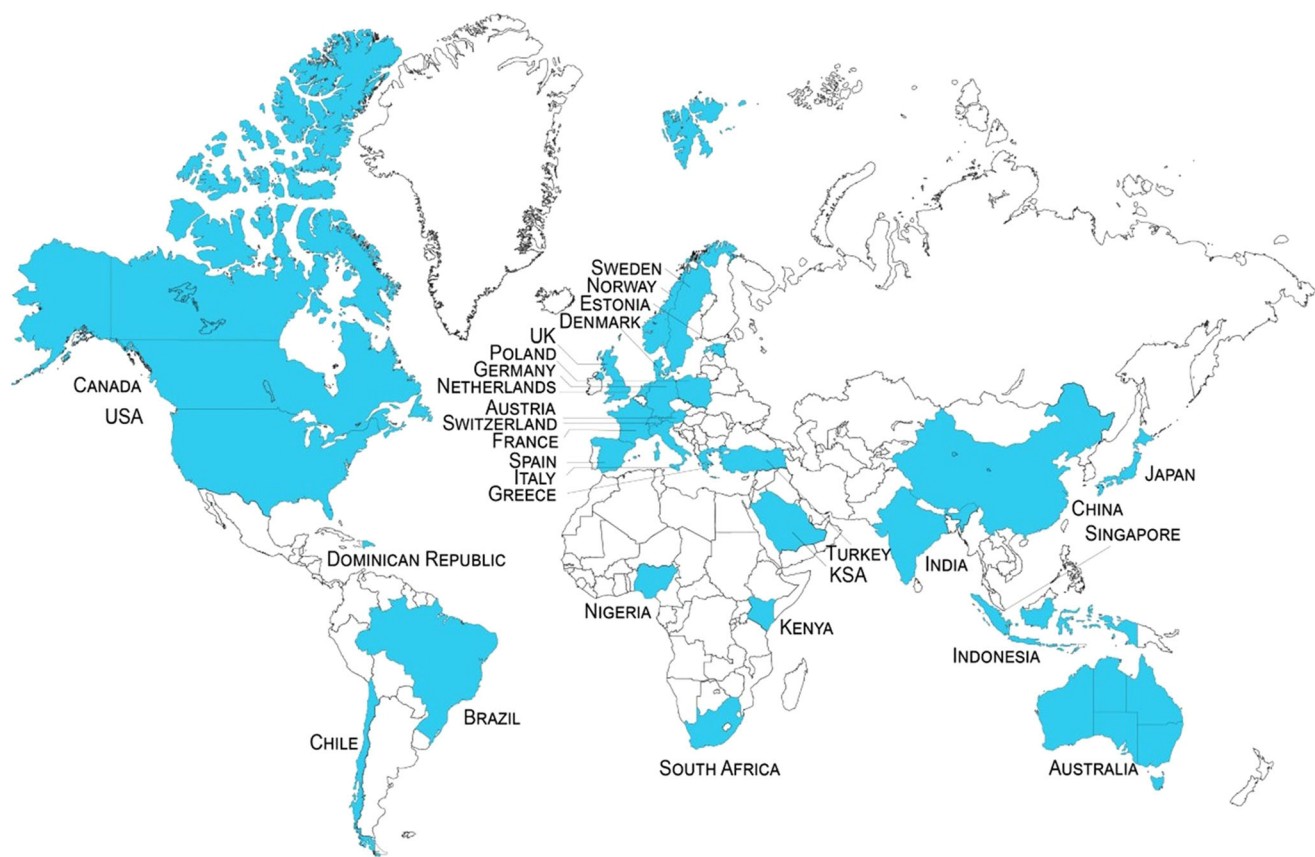

**Fig. 1 | Geographic outline of 30 countries surveyed on climate-intervention technologies.** Nationally representative surveys of the public perceptions of climate-intervention technologies were conducted in 30 countries and 19 languages (with at least 1000 participants for each country). Survey samples were nationally representative in terms of age (between 18 and 74), gender, and geographic region and with broad quotas for education and income; Source: Authors.

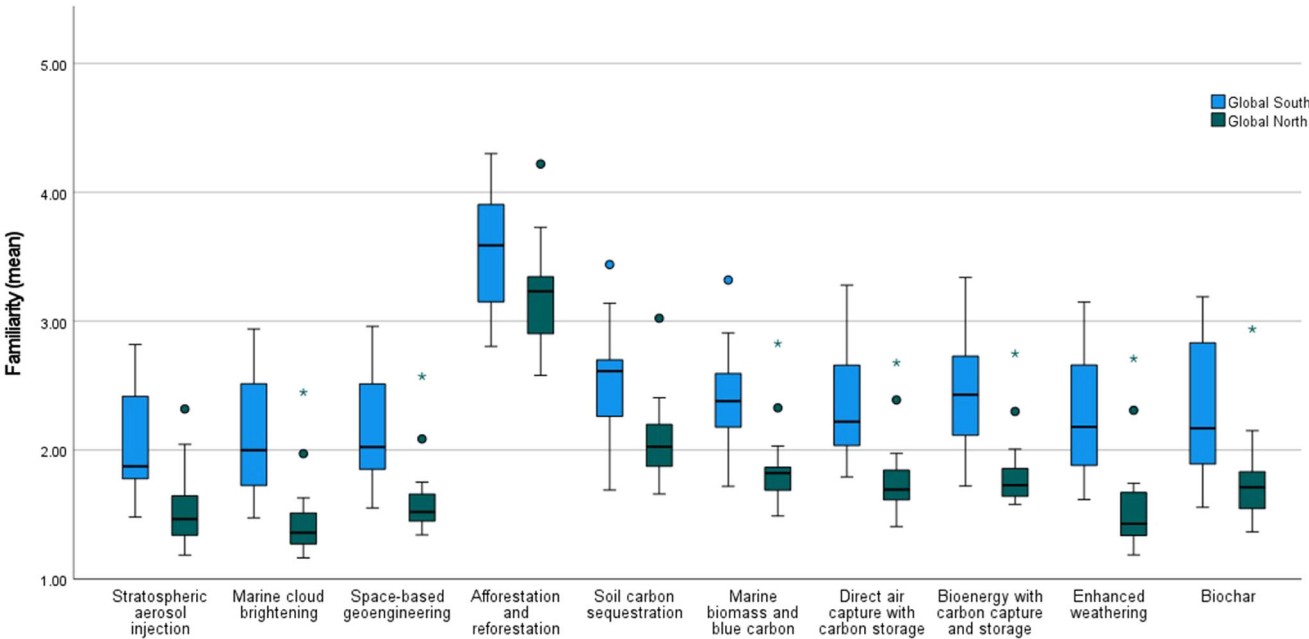

**Fig. 2 | Greater Expressed Familiarity (mean) with Climate-Intervention Technologies in Global South versus Global North countries (1–5 scale: 1 = Not at all (never heard of it); 5 = Very familiar).** $N = 30,284$ participants. Significant differences ($p < 0.05$) between expressed level of familiarity in Global North versus Global South countries were identified using nonparametric independent-samples (two-tailed) Mann Whitney $U$ tests for nine of the ten technologies: for stratospheric aerosol injection, $U = 28.000$, $Z = −3.293$, $p = 0.001$; marine cloud brightening, $U = 18.000$, $Z = −3.723$, $p = 0.000$; space-based geoengineering, $U = 26.000$, $Z = −3.378$, $p = 0.000$; marine biomass and blue carbon, $U = 33.000$, $Z = −3.077$, $p = 0.001$; soil carbon sequestration, $U = 42.000$, $Z = −2.690$, $p = 0.006$; DACCS, $U = 22.000$, $Z = −3.551$, $p = 0.000$; BECCS, $U = 23.000$, $Z = −3.507$, $p = 0.000$; enhanced weathering, $U = 22.000$, $Z = −3.551$, $p = 0.000$; and biochar, $U = 36.000$, $Z = −2.948$, $p = 0.002$. For all, the level of expressed familiarity is significantly higher for publics in Global South. The one exception is afforestation and reforestation, where there was not a statistically significant difference, although close to the threshold of significance: $U = 59.000$, $Z = −1.958$, $p = 0.052$. Horizontal lines within boxes represent medians, while shaded boxes identify the middle 50% of the data (interquartile range, IQR). Whiskers extend above and below box to either the minimum and maximum values in the sample population or 1.5 times the length of the IQR, whichever is closer to median. Shaded circles and asterisks represent outliers: with circles identifying values within 1.5 times IQR, asterisks those 1.5 to 3 times IQR. Outliers: Turkey (positive) for all technologies; USA (positive) for marine cloud brightening; space-based geoengineering; marine biomass and blue carbon, DACCS, BECCS, and enhanced weathering; and India (positive) for soil carbon sequestration and marine biomass and blue carbon.

significant across geographies and polities[26,27]. Given the potential framing effects of naturalness[12,28], we strictly avoided discussion of any approaches as being more or less natural.

## Results

This section is organized around core findings concerning the four main outcome variables of the survey: (1) familiarity, (2) risks and benefits, (3) level of support for three different types of activities (research in a laboratory setting, small-scale field trials, broad deployment), along with (4) the kinds of policies (at national and international level) that participants would deem necessary before a given technology category were used (see "Methods"). The question on policy preferences represents one of the first times that such a question has been considered in a survey on climate-intervention technologies[18]. By focusing on these variables, we establish a global baseline of public perceptions of climate-intervention technologies: how perceptions do and do not differ across publics in the Global North and Global South and the extent to which differences exist across technology categories. Given the starkly disproportionate attention to Global North in prior research, we contend that even findings of aspects where the two do not differ are meaningful, as these identify areas where the prevailing literature has a potentially greater level of generalizability.

### Familiarity with climate intervention technologies

As noted in the literature[12,14,15], familiarity with almost all climate-intervention technologies is low – this is illustrated here on a much larger scale. The only option with which participants expressed any familiarity is afforestation and reforestation; otherwise, means indicate a general unfamiliarity (Fig. 2). The technologies deemed the least familiar included SRM options and enhanced weathering, while greater familiarity is expressed with the ecosystem-based CDR approaches.

Yet, the familiarity expressed by participants is significantly higher in the Global South across all technologies, except afforestation and reforestation ($p = 0.052$). We further observe greater heterogeneity within the Global South cohort, suggesting the countries have, or rather perceive themselves as having, distinct levels of familiarity. It is however doubtful whether this signifies that those in the Global South are any more familiar with the techniques, especially given that participants in some countries consistently claim higher familiarity (Fig. 2). This tendency may reflect the influence of unobserved factors, like the differential emphasis in media and public discourse (on climate issues), or cultural tendencies, e.g., to act as an individual or express confidence in one's knowledge. For instance, a proclivity has been identified towards extreme responses in Western countries versus mid-point values in Asian ones[29,30]. As such, the higher levels of familiarity expressed may reflect responses which cluster more to the center in some countries. Of interest, the one country that is an outlier in the Global North for all technologies is Turkey, with levels of familiarity more reminiscent of those in the Global South.

### Significant differences in perceptions of risks and benefits between Global South versus Global North countries

We next examined perceptions of diverse risks and benefits to gain insights into areas that were seen to be problematic or most promising across the technology categories. These items were identified and

| | | Solar Radiation Modification | Engineered Carbon Dioxide Removal | Ecosystem-based Carbon Dioxide Removal |
|---|---|---|---|---|
| **Benefits** | | | | |
| Can be done safely in a controlled fashion | Global South | **4.88** | **5.12** | **5.62** |
| | Global North | **4.04** | **4.56** | **5.26** |
| Cost-efficient and cheaper than cutting use of fossil fuels | Global South | **4.07** | **4.49** | **5.22** |
| | Global North | **3.43** | **3.86** | **4.80** |
| Environmentally friendly | Global South | **4.83** | **5.09** | **5.73** |
| | Global North | **4.06** | **4.49** | **5.49** |
| Can be counted on in the long term | Global South | **4.47** | **4.96** | **5.51** |
| | Global North | **3.78** | **4.38** | **5.21** |
| **Risks** | | | | |
| Leads to unintended side effects | Global South | **4.25** | 4.14 | 3.70 |
| | Global North | **4.38** | 4.18 | 3.49 |
| Would distribute risks unequally between rich and poor countries | Global South | **4.23** | 4.27 | 3.97 |
| | Global North | **4.02** | 4.19 | 3.97 |
| Threat to humans and nature | Global South | **3.81** | 3.81 | **3.20** |
| | Global North | **3.90** | 3.77 | **2.87** |
| Would decrease the motivation to reduce CO2 emissions | Global South | **4.40** | **4.64** | 4.43 |
| | Global North | **4.22** | **4.16** | 4.02 |

**Fig. 3 | Significant differences between perceptions of risks and benefits in Global South versus Global North countries, grouped by technology category (bolded font indicates significant difference ($p < 0.05$) between perceptions in Global South and Global North; 1–7 scale: 1 = Strongly disagree, 4 = Neither agree nor disagree, 7 = Strongly agree; color scheme shifts from redder to greener as perceived benefits increase and perceived risks decrease).** $N = 30,284$ participants. Statistically significant differences ($p < 0.05$) between Global North and South cohorts (by technology category), according to independent-samples (two-tailed) Mann-Whitney $U$ testing. Test statistics for SRM benefits: "can be done safely in a controlled fashion", $U = 110.50$, $Z = -6.950$, $p < 0.001$; "cost-efficient and cheaper than cutting use of fossil fuels", $U = 385.50$, $Z = -4.647$, $p < 0.001$; "environmentally friendly", $U = 120.50$, $Z = -6.866$, $p < 0.001$; "can be counted on in long term", $U = 1219.50$, $Z = -6.037$, $p < 0.001$. Test statistics for SRM risks: "leads to unintended side-effects", $U = 1228.00$, $Z = 2.408$, $p = 0.016$; "threat to humans and nature", $U = 119.50$, $Z = 2.094$, $p = 0.036$; "would distribute risks unequally between rich and poor countries", $U = 608.00$, $Z = -2.785$, $p = 0.005$; "would decrease motivation to reduce CO2 emissions", $U = 680.00$, $Z = -2.182$, $p = 0.029$. Test statistics for ecosystem-based CDR benefits: "can be done safely in controlled fashion", $U = 395.00$, $Z = -4.568$, $p < 0.001$; "cost-efficient and cheaper than cutting use of fossil fuels", $U = 269.50$, $Z = -5.619$, $p < 0.001$; "can be counted on in long term", $U = 428.00$, $Z = -4.292$, $p < 0.001$ "environmentally friendly", $U = 542.00$, $Z = -3.337$, $p < 0.001$. Test statistics for ecosystem-based CDR risks: "leads to unintended side-effects", $U = 757.00$, $Z = -1.537$, $p = 0.124$; "would distribute risks unequally between rich and poor countries", $U = 963.00$, $Z = 0.188$, $p = 0.851$; "threat to humans and nature", $U = 627.00$, $Z = -2.625$, $p = 0.009$; "would decrease motivation to reduce CO2 emissions", $U = 414.50$, $Z = -4.405$, $p < 0.001$. Test statistics for engineered CDR benefits: "can be done safely in controlled fashion", $U = 350.00$, $Z = -7.200$, $p < 0.001$; "cost-efficient and cheaper than cutting use of fossil fuels", $U = 470.50$, $Z = -6.544$, $p < 0.001$; "environmentally friendly", $U = 434.00$, $Z = -6.743$, $p < 0.001$; "can be counted on in long term", $U = 374.50$, $Z = -7.067$, $p < 0.001$. Test statistics for engineered CDR risks: "leads to unintended side-effects", $U = 1755.00$, $Z = 0.452$, $p = 0.651$; "would distribute risks unequally between rich and poor countries", $U = 1487.00$, $Z = -1.088$, $p = 0.314$; "threat to humans and nature", $U = 1596.00$, $Z = -0.414$, $p = 0.679$; and "would decrease motivation to reduce CO2 emissions", $U = 469.50$, $Z = -6.549$, $p < 0.001$. Means (rather than mean ranks, on which tests are based) reported. Cells colored according to following scheme: for benefits, those above midpoint from 4.00–4.49 shaded pale yellow, from 4.50-4.99 pale green, 5.00–5.49 slightly darker green, 5.50-6.00 dark green. Conversely, cells below midpoint from 3.50-3.99 pale orange, between 3.00-3.49 orangish-red. Shading scheme inverted for risk items, as agreement here signifies a stronger sense of risks present. Cells above mid-point from 4.00-4.49 shaded pale orange, those just below pale yellow. As values decrease away from mid-point, they become greener in shade.

adapted from earlier studies[31,32], to comprehensively examine benefits and risks of economic, environmental, social, and technical or safety-related nature (Fig. 3).

Given the correspondence between technologies in a category, we focus on differences in how technology categories are perceived for this analysis: solar radiation modification; ecosystem-based CDR; engineered CDR. Looking at Fig. 3, we identify many significant differences ($p <. 05$) across the technology categories, with benefits of ecosystem-based CDR perceived to be greater, followed by engineered CDR, and then SRM. This echoes previous findings of a preference for CDR over SRM[13,31], with ecosystem-based CDR seen most favorably[32–35].

Regarding perceived benefits, almost all the values in the first four rows of Fig. 3 were above the mid-point of 4.0, indicating broad agreement that the benefits can be expected. Conversely, the level of negativity in the Global North on whether the technologies would be cost-efficient and cheaper than cutting use of fossil fuels (row 2) is noteworthy – especially for SRM. Indeed, of the three times where participants expressed disagreement that a benefit exists (i.e., the value is less than 4.0), all were in the Global North: SRM and engineered CDR both not being more cost-effective than mitigation (row 2), and not being able to count on SRM in the long term (row 4). Taken together, these results indicate substantial concerns in the Global North around the viability and cost-effectiveness of SRM and the potential adverse effects of engineered CDR on the mitigation of fossil-fuel emissions (row 4).

We fail to identify patterns for risks as clear as those for benefits – beyond risks of ecosystem-based CDR being generally lower overall for both Global North and Global South cohorts. If one assumes

engineered CDR and SRM to be more technology-centric approaches, a potential implication here is that approaches perceived to be more natural or grounded in ecosystems appear to be viewed as less risky than those relying more on (novel) technologies.

The picture is more mixed for engineered CDR and SRM, depending on the risk in question and at times varying between Global North and Global South. When gauging the potential for an unequal distribution of risk between rich and poor countries (row 6), the Global North cohort highlights engineered CDR as the most problematic category; ecosystem-based CDR and SRM are perceived to pose lesser and comparable degrees of such risk. Meanwhile, the Global South cohort perceives ecosystem-based CDR as less risky vis-à-vis unequal distributions of risks and benefits, while engineered CDR and SRM are comparably riskier. Similarly, the Global South did not distinguish between potential risks of unintended side effects for engineered CDR and SRM (row 5), but the Global North cohort identifies SRM as comparatively riskier in this regard. Finally, regarding the risks of the technology category decreasing motivation to reduce $CO_2$ emissions (row 8), the Global North cohort pinpointed SRM as posing greater risk than ecosystem-based CDR, while the Global South viewed them as equally dangerous – according to public in both the Global North and Global South, engineered CDR posed greater such risk of moral hazard than any other technology category.

Significant differences ($p < 0.05$) also emerge between Global North and Global South cohorts (Fig. 3). Comparing the values for Global North and Global South, we conclude that those in the Global South perceive greater benefits for all technology categories (rows 1–4). Though this positivity is most evident for ecosystem-based CDR, the scope of the differences between Global North and Global South is largest for SRM. While the Global North cohort is slightly negative about the benefits of SRM (values just above or below mid-point of 4.0), there is broad positivity in the Global South. While generally less positive on SRM overall, respondents across the Global South appear to be positively disposed to all technology types.

Shifting to risks (rows 5-8), the picture is more nuanced, with disagreement both in relation to technology category and the risk in question. Notably, perceptions of Global North and Global South publics significantly differ for all risks regarding SRM – but not in the same direction. Those in the Global North express greater concern around unintended side effects (row 5) and the threat to humans and nature (row 7). Meanwhile, the Global South public were more likely to highlight the unequal distribution of risks between poor and rich countries (row 6) as well as the potential to decrease motivation to reduce $CO_2$ emissions (row 8). In fact, the greater concern of engendering so-called moral hazard or mitigation deterrence extended in the Global South to both SRM and CDR. Such concern was in fact greatest for engineered CDR. Thus, the extent to which technologies are viewed as failing to address the root causes of climate change[15], or giving excuses to not reduce emissions, emerges as a crux issue[16,36]. In another study, reading about engineered CDR (or CDR in general) diminishes support for climate mitigation, by reducing the perceived threat of climate change[37] – actual evidence of moral-hazard effects at the individual level remains however mixed[13,38,39].

Two other statistically significant differences involved perceived threats to humans and nature (row 7). First, the Global South cohort comparatively emphasized such threats for ecosystem-based CDR – even if this risk was generally rated as being low. Perhaps this reflects the Global South having acted as sites for such activities, e.g., tropical, forested countries in the Reducing Emissions from Deforestation and Forest Degradation (REDD+) program, and indicates public awareness of uneven power relations and structures[40,41]. However, it must be noted the Global North and Global South did not significantly differ in their perceived risks of carbon removal for unfair burden-sharing across rich and poor countries (row 6). Also, Global North and Global South cohorts were aligned on most of the risks of engineered CDR, with the notable exception of mitigation deterrence (row 8).

## Significant differences in levels of support for climate-intervention technologies between Global North and Global South

We queried participants on their support for three kinds of activity related to each technology, roughly corresponding to stage and scale. We found via nonparametric testing (related-samples Friedman's two-way analysis of variance by ranks) that small-scale field trials tended to receive greater support than non-applied research, described as "e.g., modeling of effects on the climate and lab experiments", and broader deployment: $\chi^2(2) = 347.215$, $p = 0.000$. Pursuing broader deployment proved to be the least popular of the activities, with a mean for the global sample of 3.699 versus 3.733 for research and 3.893 small-scale field trials. Since the mean is above the scale mid-point (3), this still indicates broad openness towards such activities.

Turning to Global North versus Global South, we established, using the nonparametric related-samples Wilcoxon signed-rank test, there is higher support for research over broad deployment in the Global North, $W = 3761.00$, $Z = -6.927$, $p < 0.001$. Conversely, the Global South cohort rated the two activities equally, $W = 2831.00$, $Z = -0.180$, $p = 0.857$. Small-scale field trials received the highest support in both, according to related-samples Friedman's two-way analysis of variance by ranks: Global North, $\chi^2(2) = 251.022$, $p = 0.000$; Global South, $\chi^2(2) = 100.906$, $p = 0.000$.

We then examined differences in the level of public' support for climate-intervention technologies between the Global North and Global South by considering the composite level of support across all activities (Fig. 4; Table 1) – responses for the three types of activities were combined by taking their average, given the high degree of correlation between them. We first stress the broad preference for the ecosystem-based CDR options, with afforestation and reforestation receiving significantly greater support than other options (Fig. 4; Table 1). At the other end of the spectrum, the SRM options along with enhanced weathering received the lowest support. Lack of support for enhanced weathering is notable, especially against the relatively positive perceptions for biochar, since the two are increasingly employed together in CDR field trials on agricultural soils[42]. This may reflect concerns over the possible use of large-scale enhanced weathering in oceans, as ocean alkalinity enhancement, together with links to ocean storage, which have proven to be an issue in previous studies[15,43]. Alternatively, lack of support could be informed by problematic associations with mining[44,45], whereby enhanced weathering might run the risk of requiring greater extractive activities.

Importantly, we also identify greater support for all technologies in the Global South, except afforestation and reforestation (Table 1). Thus, afforestation and reforestation is the option that receives the most support, and the only ones for which the cohorts are in agreement. Overall, the relative preferences for climate-intervention technologies are analogous for the Global North and Global South cohorts. Ecosystem-based CDR approaches stand apart at the top in both, with afforestation and reforestation in a category of its own. Also, the SRM options (stratospheric aerosol injection in particular) and enhanced weathering jointly have the least support. Where the cohorts differ is how engineered CDR approaches are perceived, i.e., the extent to which they are viewed similarly to SRM options. For instance, there is similar support in the Global South cohort for engineered CDR methods and marine cloud brightening. While there is no blurring between SRM and engineered CDR (beyond enhanced weathering) in the Global North cohort, there emerges greater support for biochar, so that it is separated from DACCS and BECCS and placed on its own. In this way, there is some evidence for how the cohorts vary in terms of their relative preferences for engineered CDR.

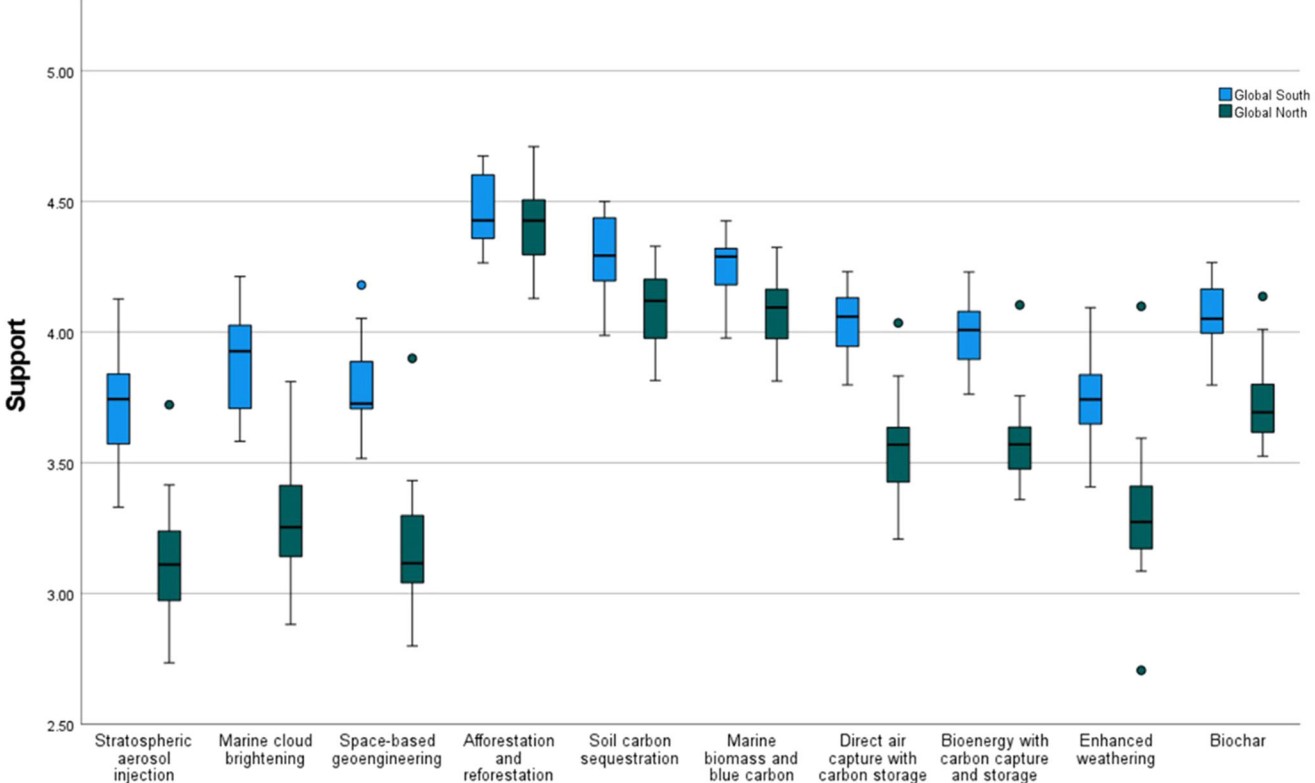

**Fig. 4 | Significant Differences in Support (Research, Small-Scale Field Trials, and Broad Deployment) for Climate-Intervention Technologies in Global South versus Global North countries (1-5 scale: 1 = Strictly reject; 3 = Neither support nor reject; 5 = Fully support).** $N = 30{,}284$ participants. Support refers to overall support for technology activities in terms of research, small-scale field trials, and broad deployment. Horizontal lines within boxes represent the median, while the shaded box identifies the middle 50% of the data (the interquartile range, or IQR). Whiskers extend above and below the box to the minimum and maximum values in the sample population or to 1.5 times the length of the IQR, whichever is closer to median. Outliers include: Turkey (positive) for stratospheric aerosol injection, space-based geoengineering, DACCS, BECCS, enhanced weathering, biochar; Italy (negative) for enhanced weathering; and India (positive) for space-based geoengineering.

**Table 1 | Significant Differences ($p < 0.001$) in means of age and climate change beliefs between Global North and Global South**

| | Global North (N = 19,201) | Global South (N = 11,083) | Significant difference (*) between Global North and South |
|---|---|---|---|
| Age (mean years) | 45.07 | 36.97 | * |
| Concern over climate change (1-5 scale: 5=Extremely worried) | 3.644 | 4.069 | * |
| Perceived climate harm (1-4 scale: 4 = A great deal) | 2.772 | 3.253 | * |
| Personal experience with a major natural disaster in last three years (percentage answering "Yes") | 33.28% | 53.35% | * |
| Belief in science and technology as a solution to climate change (1-5 scale: 5 = Strongly agree) | 3.337 | 3.810 | * |

Note: N = 30,284 participants. Significant differences ($p < 0.001$) between Global North and Global South cohorts, according to (two-tailed) independent-samples Mann Whitney U-testing. *Age (mean)* higher in Global North, $U = 20700.00$, $Z = 14.16$, $p = 0.000$; *concern over climate change*, $U = 3050.00$, $Z = -10.229$, $p = 0.000$, *perceived climate harm*, $U = 2000.00$, $Z = -11.68$, $p = 0.000$, *personal experience with major natural disaster*, $U = 3500.00$, $Z = -9.60$, $p = 0.000$, *science and technology as solution to climate change*, $U = 1850.00$, $Z = -11.89$, $p = 0.000$, higher in Global South. Means (not mean ranks, on which tests are based) reported.

Rather than leaving these as unexplained cohort differences, we examined how cohorts varied by climate beliefs and age (Fig. 5). Significant differences emerged, with those in the Global South younger, in accordance with the overall age of these populations. They were also more concerned about climate change and more likely to expect greater personal harm and to have experienced a major natural disaster (e.g., flood, heatwave, wildfire, blizzard) in the last three years. In total, these factors suggest a more refined account of how and why the Global South and Global North significantly differ in their support for climate-intervention technologies.

To better understand what underlies the differential support between Global North and Global South, we performed hierarchical (linear) regression analysis, where the factors of interest (i.e., age and climate change beliefs) are entered sequentially to investigate how much these account for the varying support across the two cohorts (Supporting Information, Supplementary Fig. 3). The mean age of a country's population stands out in terms of its explanatory power for support – once age is included, whether a country is in the Global North or Global South is no longer significant, indicating that this factor accounts for much of the difference. Age also emerges as differentially relevant across the technology types. While (mean) age does not have explanatory power for the support of ecosystem-based CDR, $F(1,87) = 1.290$, $p = 0.259$, Cohen's $f^2 = 0.015$ (small effect size), it does statistically explain support for engineered CDR, $F(1,117) = 15.026$,

| | Global North (N=19,201) | Global South (N=11,083) | Significant difference in Support (*) |
|---|---|---|---|
| Stratospheric aerosol injection | **3.11** | **3.70** | * |
| Marine cloud brightening | **3.27** | **3.89** | * |
| Space-based geoengineering | **3.17** | **3.80** | * |
| Afforestation and reforestation | 4.42 | 4.47 | |
| Soil carbon sequestration | **4.10** | **4.30** | * |
| Marine biomass and blue carbon | **4.07** | **4.24** | * |
| DACCS | **3.55** | **4.04** | * |
| BECCS | **3.58** | **3.99** | * |
| Enhanced weathering | **3.31** | **3.76** | * |
| Biochar | **3.73** | **4.06** | * |

**Fig. 5 | Greater Support for Climate Intervention Technologies in Global South versus Global North Countries (1-5 scale: 1 = Strictly reject; 3 = Neither support nor reject; 5 = Fully support; bolded font indicates significant difference ($p < 0.05$) between Global South and Global North; color scheme shifts from redder to greener as support for technology increases).** $N = 30,284$ participants. Statistically significant differences ($p < 0.05$) of level of technology support between Global North and Global South cohorts, according to nonparametric independent-samples (two-tailed) Mann-Whitney $U$ testing. We identified significantly greater support in the Global South for nine of the ten technologies: stratospheric aerosol injection, $U = 9.00$, $Z = −4.110$, $p = 0.000$; marine cloud brightening, $U = 4.00$, $Z = −4.325$, $p = 0.000$; space-based geoengineering, $U = 8.00$, $Z = −4.153$, $p = 0.000$; soil carbon sequestration, $U = 37.00$, $Z = −2.905$, $p = 0.003$; marine biomass and blue carbon, $U = 39.00$, $Z = −2.819$, $p = 0.004$; DACCS, $U = 7.00$, $Z = −4.196$, $p = 0.000$; BECCS, $U = 9.00$, $Z = −4.110$, $p = 0.000$; enhanced weathering, $U = 15.00$, $Z = −3.852$, $p = 0.000$; biochar, $U = 13.00$, $Z = −3.938$, $p = 0.000$. The one exception is afforestation and reforestation, $U = 88.00$, $Z = −0.710$, $p = 0.497$. Means (rather than mean ranks, on which tests are based) reported. Support refers to overall support for activities vis-a-vis research, small-scale field trials, broad deployment. Cells colored according to following scheme: if support for technology is between 3.00–3.40, we applied pale orange; from 3.41–3.80, pale yellow; from 3.81–4.20, pale green; above 4.20, a darker green.

$p < 0.001$, $f^2 = 0.128$ (medium effect size) and SRM, $F(1,87) = 40.423$, $p < 0.001$, $f^2 = 0.466$ (large effect size) (see Supplementary Fig. 5). However, while the effect size for SRM is large, that for engineered CDR is small in nature. There are initial indications that it is thus not necessarily whether a country is in the Global North or Global South but rather the youthfulness of the population that correlates to the level of support for climate-intervention technologies.

## Public support for national and international policy options of climate-intervention technologies

Lastly, public preferences vis-à-vis support for policies or governance activities were assessed by asking participants which options they would deem necessary before technologies would be used. As opinions and preferences of the public will be influential for future policy discussions, whether in a consultation role or by refusing the social license to operate or engaging in protest, it is also important to gain insights in this regard. Policy support was assessed at a category level: for SRM, engineered CDR, ecosystem-based CDR. Participants were presented the same set of options (Fig. 6), from which they could choose as many as they desired or "None of the above". Regarding the option set, there was one difference between categories: for CDR, individuals could select "Global-level market for trading carbon credits and/or offsets"; for SRM, this was substituted by "Creation of new international scientific agency to explore different technologies and start testing and development".

Looking at Fig. 6, we identify significantly higher support for most policies across the Global South. Support for policies thus mirrors the support for technologies. Comparing the respective mean percentages, support for many policies is ten percentage-points greater (or more) than in the Global North – notably, for national-level support and funding by governments (row 2) and most activities undertaken at the international level (rows 5-8). Overall, this signals there is more trust in international organizations and related activities across the Global South. Indeed, the only option generally preferred by the Global North is "None of the above" (row 9). Those in the Global

South instead were more broadly supportive of policies related to CDR and SRM.

Furthermore, we identify several commonalities on a global level regarding the least and most popular policies for SRM and CDR. Policies receiving the greatest support across all categories are national-level support and funding by governments (row 2) and information campaigns to consult and inform public (row 3). In contrast, the least supported options were independent national government restrictions (row 1) and international bans or moratoria on technologies deemed risky (row 4). Indeed, support for a more restrictive approach, whether at the national or international level, is relatively absent for all technology categories, suggesting there might be a limited public interest in the kind of proposals debated in academic or policy circles[7,46]. Participants in Global North and Global South are also aligned on the importance of an international organization for oversight and setting standards on ecosystem-based CDR (row 5, column 3).

Also, the CDR-specific option of a global-level carbon market receives limited support for both ecosystem-based and engineered options (row 7). Occupying the intermediate layer of support, there are international-level options, like generating a special report (e.g. by IPCC) to evaluate and assess technologies (row 8) or establishing an international organization for oversight and standard setting (row 5). As such, there emerges a clear preference (in the Global South and Global North) for national-level policies that are more supportive and information-focused activities. More restrictive approaches, whether at the national or international level, are less preferred.

What is also apparent is how similar policy preferences prove to be across technology categories (Fig. 6) – as a reminder, participants were randomly assigned to three groups and thus responses are independent of one another. Participants only assigned differing support by technology category for three of the policy options (Table 2). This entailed greater support for national-level support and funding for private and public R&D for ecosystem-based CDR over SRM in the Global North and Global South (rows 1-2) – with engineered CDR given an intermediate position in the Global North. This is the

|  |  | SRM | Engineered CDR | Ecosystem-based CDR |
|---|---|---|---|---|
| *Domestic* |  |  |  |  |
| Independent national government policies restricting use of technology | Global South | **34.30%** | **35.97%** | **33.15%** |
|  | Global North | **26.74%** | **26.14%** | **24.91%** |
| National support and funding for public and private R&D | Global South | **60.85%** | **64.80%** | **67.66%** |
|  | Global North | **44.52%** | **53.75%** | **62.11%** |
| Information and engagement campaigns to consult / inform public | Global South | **60.85%** | **60.42%** | **62.86%** |
|  | Global North | **49.62%** | **51.96%** | **53.13%** |
| *International* |  |  |  |  |
| International ban or moratorium on technologies deemed risky | Global South | 33.58% | 35.95% | 34.35% |
|  | Global North | 35.49% | 35.22% | 34.06% |
| International organization to conduct oversight and set standards | Global South | **54.31%** | **56.04%** | 56.69% |
|  | Global North | **43.20%** | **47.40%** | 50.82% |
| Scientific agency to explore and start testing and development | Global South | **53.93%** | . | . |
|  | Global North | **44.11%** |  |  |
| Global-level market for trading carbon credits and/or offsets | Global South | . | **34.85%** | **35.80%** |
|  | Global North |  | **24.23%** | **26.49%** |
| Special report (e.g., by IPCC) to evaluate and assess technologies | Global South | **53.22%** | **55.06%** | **56.89%** |
|  | Global North | **40.18%** | **44.50%** | **44.99%** |
| None of the above | Global South | **3.05%** | **3.67%** | **3.35%** |
|  | Global North | **12.44%** | **11.97%** | **9.43%** |

**Fig. 6 | Differences in Policy Support between Global North and Global South, assessed by technology category (difference in percentage of respondents selecting policy option; bolded font indicates significant difference ($p < 0.05$) between Global South and Global North; color scheme shifts from redder to greener as percentage selecting option increases; SRM stands for Solar Radiation Modification, CDR for Carbon Dioxide Removal).** $N = 30,284$ participants. Statistically significant differences ($p < 0.05$) in *p*ercentage of respondents expressing support for policy options in Global South versus Global North, according to nonparametric independent-samples (two-tailed) Mann-Whitney *U* tests. Here, the test statistics for SRM: "independent national government policies restricting use of technology", $U = 35.50$, $Z = -2.970$, $p = 0.002$; "national support and funding to enable technology for public and private R&D", $U = 11.50$, $Z = -4.004$, $p = 0.000$; "information and engagement campaigns to consult and inform public", $U = 28.00$, $Z = -3.293$, $p = 0.001$; "international ban or moratorium on technologies deemed risky", $U = 129.00$, $Z = 1.055$, $p = 0.307$; "international organization to conduct oversight and set standards", $U = 20.50$, $Z = -3.615$, $p = 0.000$; "scientific agency to explore and start testing and development", $U = 19.00$, $Z = -3.680$, $p = 0.000$; "special report (e.g., by IPCC) to evaluate and assess technologies", $U = 0.00$, $Z = -4.498$, $p = 0.000$; "none of the above, $U = 205.50$, $Z = 4.348$, $p = 0.000$. The test statistics for engineered CDR: "independent national government policies restricting use of technology", $U = 27.00$, $Z = -3.336$, $p = 0.000$; "national support and funding to enable technology for public and private R&D", $U = 11.00$, $Z = -4.025$, $p = 0.000$; "information and engagement campaigns to consult and inform public", $U = 46.50$, $Z = -2.497$, $p = 0.011$; "international ban or moratorium on technologies deemed risky", $U = 97.00$, $Z = -0.323$, $p = 0.767$; "international organization to conduct oversight and set standards", $U = 44.00$, $Z = -2.605$, $p = 0.008$; "global-level market for trading carbon credits and/or offsets", $U = 22.00$, $Z = -3.551$, $p = 0.000$; "special report (e.g., by IPCC) to evaluate and assess technologies", $U = 21.50$, $Z = -3.572$, $p = 0.000$; or "none of the above, $U = 200.50$, $Z = 4.132$, $p = 0.000$. Test statistics for ecosystem-based CDR: "independent national government policies restricting use of technology", $U = 43.00$, $Z = -2.647$, $p = 0.007$; "national support and funding to enable technology for public and private R&D", $U = 58.00$, $Z = -2.002$, $p = 0.047$; "information and engagement campaigns to consult and inform public", $U = 37.00$, $Z = -2.905$, $p = 0.003$; "international ban or moratorium on technologies deemed risky", $U = 99.50$, $Z = -0.215$, $p = 0.832$; "international organization to conduct oversight and set standards", $U = 64.00$, $Z = -1.743$, $p = 0.085$; "global-level market for trading carbon credits and/or offsets", $U = 28.00$, $Z = -3.293$, $p = 0.001$; "special report (e.g., by IPCC) to evaluate and assess technologies", $U = 9.50$, $Z = -4.089$, $p = 0.000$; or "none of the above, $U = 199.00$, $Z = 4.068$, $p = 0.000$. Cells colored according to following scheme: if percentage selecting option is between 0–20%, red is used; 20-40%, pale orange; 40–60%, pale yellow; 60% or more, green.

only distinction related to the technology category in the Global South – and the sole distinction made between types of CDR. Publics in the Global North also expressed greater support for establishing an international organization to conduct oversight of CDR versus SRM (row 3) and for a special report at the international level (e.g., by IPCC) for CDR (row 4). We thereby interpret a slightly greater tendency to distinguish between technology categories in the Global North. Where such differences are made, we find that policy options related to SRM generally receive less support. In fact, while information and engagement campaigns receive the second-highest support in the Global North, the decline in support relative to the Global South amounts to only a (near) majority in the case of SRM.

## Discussion

This paper reports the results of the first global-level, cross-country survey of public perceptions and support for CDR and SRM. Given its scale (i.e., 30 countries, with a broad representation of the Global South and all regions) and scope (ten different technologies, with a joint evaluation of three to four options), the survey provides an initial global baseline for public perceptions, differing by technology category and of Global North versus Global South. As with all research into public perceptions of novel, swiftly-evolving subjects, survey results should be interpreted with caution – and could represent uninformed "pseudo-opinions"[47] or prove highly malleable to subsequent information and experience[48].

Given that the public remains broadly unfamiliar with these techniques (except for afforestation and reforestation) and political and civic discussions are generally nascent, exercises that center the perceptions and opinions of the public, even when these could be misinformed, are critical for advancing our understanding of the status quo. We moreover contend that a survey with the current scope, scale, and inclusiveness of the Global South provides crucial insights into

**Table 2 | Policy Options with Significant Differences in Support by Technology Category (p < 0.05) (difference in percentage of respondents opting for policy option; SRM stands for Solar Radiation Modification, CDR for Carbon Dioxide Removal)**

| | SRM (1) | Significant difference (1) vs. (2) | Engineered CDR (2) | Significant difference (2) vs. (3) | Ecosystem-based CDR (3) | Significant difference (1) vs. (3) |
|---|---|---|---|---|---|---|
| **Global South** | | | | | | |
| National support and funding for public and private R&D | 60.85% | | 64.80% | | 67.66% | * |
| **Global North** | | | | | | |
| National support and funding for public and private R&D | 44.52% | * | 53.75% | * | 62.11% | * |
| International organization to conduct oversight and set standards | 43.20% | * | 47.40% | * | 50.82% | * |
| Special report at international level (e.g., by IPCC) to evaluate and assess technologies | 40.18% | * | 44.50% | * | 44.99% | * |

N = 30,284 participants. Statistically significant differences (p < 0.05) in percentage of respondents expressing support for policy options across technology categories, according to pairwise, independent-samples (two-tailed) Kruskal-Wallis H testing, with significance values adjusted using Bonferroni correction for multiple tests. Test statistics for Global South: "national support and funding for public and private R&D", SRM vs. ecosystem-based CDR, H = −12.136, Z = −2.944, p = 0.010, SRM vs. engineered CDR, H = −6.136, Z = −1.489, p = 0.410, ecosystem-based vs. engineered CDR, H = 6.000, Z = 1.456, p = 0.436. Test statistics for Global North: for "national support and funding for public and private R&D", SRM vs. ecosystem-based CDR, H = −28.368, Z = −5.268, p = 0.000, SRM vs. engineered CDR, H = −14.737, Z = −2.737, p = 0.019; ecosystem-based vs. engineered CDR, H = 13.632, Z = 2.532, p = 0.034; for "international organization to conduct oversight and set standards", SRM vs. ecosystem-based CDR, H = −21.921, Z = −4.072, p = 0.000, SRM vs. engineered CDR, H = −14.079, Z = −2.615, p = 0.027, ecosystem-based vs. engineered CDR, H = 7.842, Z = 1.457, p = 0.145; for "special report at international level (e.g., by IPCC) to evaluate and assess technologies", SRM vs. ecosystem-based CDR, U = 260.00, Z = 2.321, p = 0.020, SRM vs. engineered CDR, U = 255.50, Z = 2.190, p = 0.027; ecosystem-based vs. engineered CDR, U = 175.00, Z = −0.161, p = 0.885. Independent-samples Mann-Whitney U testing was used for latter tests to resolve discrepancies in the significance findings, and given the conservative tendencies of Bonferroni correction[72,73]. Further details on test statistics, significance testing, exact p-values in Supporting Information (see also Supplementary Table 5a and b).

where public perceptions of these technologies are at present. We thereby draw several conclusions about the preferences of the public around the world, including those broadly held in common, and those crucial areas of disagreement.

For one, echoing other studies in the literature[14,32–35,49,50], those approaches perceived to be more natural, e.g., afforestation and soil carbon sequestration, are broadly preferred and viewed as having a better balance of benefits to risks. This carries over into greater support for CDR vis-à-vis SRM, though there are notable exceptions. Although there is broad positivity towards ecosystem-based CDR approaches, support for engineered approaches was more discrepant. Perhaps most strikingly, enhanced weathering is perceived (in terms of support and risk-benefit balance) as the least desirable CDR approach and is even rated similarly to one SRM approach (marine cloud brightening). This could, in part, have to do with a lack of familiarity[19], but it is quite plausible that such a lack of support stems from problematic associations with increasing mining activities and the need for interventions and storage in the ocean[15,43,44].

Overall, the divergence in perceptions across CDR technologies highlights the need for caution when speaking of CDR as a coherent category. While many studies have examined perceptions of CDR in general (or SRM in general), our results underscore the value of a granular approach that examines perceptions at a technology level[15,33–35,51].

In addition, we find the scale and type of activity matter, with small-scale field trials receiving more support than research or broader deployment – indeed, with the latter two often receiving equal levels of support. This may suggest, for CDR and SRM, that public are contemplating the need for a more advanced degree of feasibility and risk assessment to take place in the real world – but with deployment still seen as the least popular of the three options, there are implied limits to how much experimentation seems desirable. On this point, our study does not consider the potentially key differences in the scale and intent between kinds of field tests[52], which may also affect public perceptions.

We also offer comprehensive insights on the similarities and differences regarding perceptions and support across the Global North and Global South. Publics in the Global South expressed a belief that there are greater benefits for all technology categories and, except for afforestation and reforestation, expressed greater support for all technologies. Those in the Global South also seemed to have a greater willingness to explore options that the Global North comparatively dismisses as unpalatable, notably SRM. Surprisingly, our survey provides evidence that Global South countries are more supportive of SRM technologies than Global North countries, viewed more benefits to research and deployment, and express stronger interest in policies that would facilitate continuing research, development, and oversight.

In terms of how the Global North and Global South differed, we found that those in the Global South expressed significantly higher levels of concern over climate change, perceived that they will experience more harm from climate change in the future, had more substantial experience recently with major natural disasters and, moreover, view science and technology as a potential climate solution. Among other things, the findings confirm and extend prior studies[53,54] on the general influence of climate harm and perceived vulnerability on the support for climate policy and action – here articulated in terms of CDR and SRM and framed on a global level. Also, at least in this respect, there is agreement between experts (i.e., negotiators and scientists involved in international climate policymaking) and the public when it comes to the motivating influence of climate beliefs[4].

Of interest, the distinguishing feature between Global North and Global South that emerges as most significant is the (mean) age of a country's population. Those living in a country with a younger population, spanning a continuum of Global South and Global North, tend to express significantly greater support for climate-intervention

technologies: engineered CDR and SRM in specific. Though age only draws a rough distinction between countries, the fact that the mean age of country population is significantly and negatively correlated with climate change beliefs points to one interesting vein for future research.

Furthermore, the Global North and Global South cohorts vary in terms of the types of risks that are most salient, with greater concern about environmental and safety risks in the Global North versus power imbalances and moral-hazard issues in the Global South. To the latter point, the Global South was more generally concerned about climate-intervention technologies lowering the motivation to reduce CO2 emissions. Research is therefore urgently needed which focuses on perceptions of moral hazard and mitigation deterrence in the Global South, not least as it is individuals in such countries most likely to bear the brunt of such hazards.

As CDR and, to a lesser extent, SRM are assigned a greater role in national and international climate discussions[3,55–57], it becomes critical to gain insight into the kinds of policies which the public would support. Previously, discussions of the policy options have relied on in-depth deliberative workshops, although principally focusing on CDR[15,58,59]. Helpfully, such research highlights the importance of socio-cultural and geographical contexts, along with the need to develop suitable governance arrangements. Insofar as this research has mostly been undertaken in a single-country context, there is a crucial opportunity to use the current results to examine policy preferences through a broader lens, contrasting by technology category and between Global North and Global South.

At a general level, we identify a clear hierarchy wherein supportive policies at the national level are most favored by respondents, including those with informational focus, followed by those aiming to foster international-level architecture. Conversely, policies that are more restrictive or limiting in nature broadly received the least support. There is comparatively limited interest now – at least when presented in a general manner – in the kinds of bans or moratoria, on SRM, discussed in the literature[7]. Similarly, the lack of support for global-level markets to trade carbon credits and offsets – preferred by under 30% of participants – is notable in view of the prominence of related negotiations at IPCC level[60,61]. Further analysis is needed to explore this disconnect between conversations taking place within expert, NGO, and decision-making circles vis-à-vis a wide array of national publics.

Drawing on the broad similarities in policy support across technology categories, we also infer a public preference for a rather coherent climate-intervention approach. Exceptions are notable, though, particularly in relation to national-level support and funding by governments for public and private R&D. While one of the most popular options overall, it is viewed as more desirable for ecosystem-based CDR. Likewise, though a large segment of participants expressed interest in an international organization devoted to oversight of all technology categories, such interest is higher for CDR. It can be surmised that support for such an organization indicates that carbon removal is seen as closer to real-world deployment, making such oversight and standards more relevant. In any case, a crucial qualification here is that policy preferences (and indeed support) have the potential to be quite heterogenous, both within and between countries.

In sum, perspectives and preferences of experts, whether in the Global North or Global South, do not necessarily reflect those of the publics across the Global South – neither in terms of their prospective support for climate-intervention technologies (which seems to be higher than that of corresponding publics in the Global North) nor for their support for more restrictive policies including bans or moratoria on technologies deemed risky. We have discussed several potential reasons for higher support among publics in the Global South, including stronger perceptions and salient experiences with climate harm. At the same time, there is an urgent need for deeper, more focused investigations of public perceptions of climate-intervention technologies in these contexts. The present unfamiliarity with many options, and the potential for perceptions to evolve, here insists on the importance of prompt engagement – not a reason to dismiss findings. The takeaway is clear: there is no substitute for directly engaging with representative publics in the Global South. Future research and endeavors are urgently needed in this vein, especially given the often-dramatic differences in public perceptions and support of climate-intervention technologies in the Global South and Global North.

## Methods
All components of the research were granted ethical approval by relevant authorities at Aarhus University (#2021–13). Full and informed consent was given by all participants before the beginning of the study, along with all participants being notified about the fact that their data would be handled in a fully anonymous manner and in complete accordance with the General Data Protection Regulation and any other pertinent data-security regulations, that any data would be analyzed in an aggregate fashion and would not be personally identifiable in any way, and that they had the right to withdraw their participation at any time. In addition, any questions about particular data being sensitive, including those that emerged in the course of the survey(s), were handled by erring on the side of caution and not asking a question in a given market. For instance, from the outset, we decided not to ask about political views in China and the question of whether one self-identified as a member of an ethnic minority or indigenous group was removed in Estonia following feedback from participants. Upon successful completion of the survey, participants received monetary compensation directly from the professional survey firm, Norstat.

### Sample design
The survey was conducted online, with alternate templates specifically designed for handheld mobile device and desktop use, in a total of 30 countries and 19 languages (see Supplementary Table 1), and a total of $N = 30,284$ participants. The countries were selected in accordance with several criteria, specifically, to have representation of the Global South, which is so far lacking in the literature, as well as all regions of the world, given the omission of Latin America and the Caribbean as well as the Middle East and Africa so far, and to ensure inclusion of some small island developing states, given the salience of severe climate threats in these countries. We also strove to include countries where research, trialling, and/or deployment into one or more climate-intervention technologies was ongoing. Our definition of Global South follows the classification of the United Nations' Finance Center for South-South Cooperation, officially used by the UN for distributing development funding. Our sample of countries (Fig. 1) includes 11 from the Global South (Brazil, Chile, India, Nigeria, Indonesia, South Africa, Kenya, Saudi Arabia, Dominican Republic, China, and Singapore), and 19 from the Global North (USA, Canada, Australia, Japan, Austria, Germany, United Kingdom, France, Sweden, Poland, Switzerland, Greece, Italy, Netherlands, Norway, Spain, Denmark, Estonia, Turkey). There are representatives on each continent, including, for the first time, several in Africa and South America and small island states, one a developing country in Latin America (Dominican Republic).

Each survey was nationally representative (with at least $N = 1000$ for each country) and was conducted in total from August to December 2022. Data collection was administered online by Norstat on behalf of Aarhus University using quota sampling, with informed consent obtained from all participants included in the study, all of whom were notified in advance of how their data would be handled as well as their right to withdraw at any time, and with all data delivered to researchers de-identified and anonymized by Norstat. Surveys were nationally representative of country populations 18–74 in terms of age, gender, and geographic region along with broad quotas for education and income. The only departures from national representativeness were: in

Singapore, for those 55–74; in Greece, Switzerland, India, Dominican Republic, Chile, India, Saudi Arabia, and Indonesia, for those 65–74; in Saudi Arabia, Indonesia, Chile, for some living in smaller regions; and in the United Kingdom, for those in North East England. In such cases, these groups were still broadly represented, albeit at levels falling short of being national representative. Demographic characteristics for countries are available on request from authors.

The survey investigated public perceptions of climate-intervention technologies by means of measures related to risks and benefits, support for different technology-related activities, support for various policy options along with questions on sociodemographic characteristics, alongside covariates such as beliefs about climate change and environment, trust in actors and institutions, aversion to tampering with nature, and credibility of sources of information (see Supplementary Table 2 in Supporting Information for full overview of survey procedure). Survey design, especially for the information texts that were provided to participants, was revised and finalized by extensive piloting, input from external researchers, professional translators, and programmers and survey experts at Norstat, as well as drawing on soft-launch results – many of the revisions are explained in the *Languages and Translation* section in Supporting Information.

### Survey measures

**Familiarity.** We provided individuals with a short information text to read on each of the three (or four) technologies in the group to which they were assigned (Supplementary Fig. 2). Following the information texts and two comprehension checks, we assessed familiarity by the question, "Before today, how familiar were you with each of the following technologies?" Responses were on a five-point scale, from 1= "not at all (never heard of it)" to 5 = "very familiar". This question was asked after rather than prior to the information texts, in view of the potential for misunderstanding about the technologies and in view of the widespread lack of familiarity on climate-intervention technologies.

**Perceptions of risks and benefits.** Eight risk and benefit perception items, four of each, are broadly adapted from earlier studies[31,32].
- …can be done safely in a controlled fashion.
- …is cost-efficient and also cheaper than cutting the use of fossil fuels.
- …is environmentally friendly.
- …can be counted on in the long term.
- …leads to unintended side effects.
- …would distribute risks unequally between rich and poor countries.
- …is a threat to humans and nature.
- …would decrease the motivation to reduce CO2 emissions.

Respondents were asked to what extent they agreed or disagreed with the following statements, each being the second half of a sentence beginning with "The deployment of this technology", with the name of the technology and its accompanying graphic (used in the information text) below, along with a hyperlink to the information. Responses were on a balanced seven-point scale: 1=Strongly disagree; 4 = Neither agree nor disagree; 7 = Strongly agree). The presentation of items was randomized with four appearing on each page. We opted against the inclusion of "Don't know" as an answer option for these items given the potential insights to be gained from nudging survey participants to take a stance one way or the other. In contrast, over-use of such an approach can encourage participants to always take this answer. Though we do include "Don't know" as an option elsewhere in the survey, we deemed it useful and informative to not afford individuals this choice for risk-benefit items.

In finalizing the set of items, we made several changes from the foregoing literature, including removing "…negatively affects the surrounding area" due to overlap with other items, adding "…is driven more by profit than the public interest" (from Cox et al.[15]). Two other items were also added, which have been used in prior research[15,16] to explore moral-hazard issues: "… would help buy more time to reduce CO2 emissions" and "…would decrease the motivation to reduce CO2 emissions". These items have been used in prior research[15,16] to explore moral-hazard issues. In addition, we removed the new item included by Jobin and Siegrist[32], "…is reversible", as this was deemed likely to be a question where participants may just echo back whatever information they were given. Based on results from the soft launch, and in particular to reduce overlap between items and to address confusion and uncertainty on the part of participants, we ultimately cut two items: "… would help buy more time to reduce CO2 emissions", as this was deemed unlikely to vary much across technologies; and "…is driven more by profit than the public interest", as this was deemed less important to consider than the other risks. The final set of eight risk and benefit perception items therefore encompasses a range of potential issues, e.g., safety, equity, cost, as well as environmental, social, and economic dimensions.

**Weighing the risks against the benefits.** We asked participants to provide a summary evaluation of the various technologies through the question, "Which of these statements best reflects your own view on the balance of the potential risks and benefits (for each of the technologies)?" The technologies were presented in random order, with the name of the technology (as a hyperlink to the information text) and accompanying graphics appearing above the question. Responses were on a five-point scale, with options of 1 = Risks far outweigh the benefits; 2 = Risks slightly outweigh the benefits; 3 = Benefits and risks are about the same; 4=Benefits slightly outweigh the risks; 5 = Benefits far outweigh the risks. This item was adapted from Pidgeon and Spence[62], to have a summary counterpart to the detailed risk and benefit perception items. In the end, given the broad correlation (Spearman's $\rho = 0.936$) between this item and the measure(s) for support, we focus on the latter in this study along with the detailed breakdown of risks and benefits in the current study.

**Support for Technology.** We investigated participants support for different activities related to carbon removal and solar radiation modification through three questions: "How much do you support further research (e.g., through modeling of effects on the climate and lab experiments) regarding each of the following technologies?"; "How much do you support small-scale field trials for each of the technologies?"; and "How much do you support the broader deployment of each of the technologies to limit the effects of climate change?" The order of these three items was always the same, while the order of presentation of technologies was randomized (though consistent across the three questions) - with the name of the technology (as a hyperlink to the information text) and accompanying graphic always appearing with the question(s). Responses were on a five-point scale, with options of: 1 = Strictly reject; 2=Somewhat reject; 3 = Neither reject nor support; 4 = Somewhat support; 5 = Fully support; along with a "Don't know" option. These questions were adapted from prior research[32,62], where we added details on what further research entailed, drawing on the examples of other research, "(e.g., through theoretical modeling and lab experiments)", along with a new item that distinguished support of small-scale field trials, i.e., as a meso-level between research and broad deployment.

Given that the three support measures were strongly correlated (i.e., the lowest Spearman's rho correlation was 0.956 between small-scale field trials and broader deployment), we constructed a composite measure for support by taking the average of them. From a principal component analysis (varimax rotation), a one-factor solution was obtained for each technology. Reliability was more than sufficient, with values of Cronbach's α for all technologies > 0.90.

**Support for policy.** To understand which policy options related to carbon dioxide removal or solar radiation management individuals would support, we asked, for each category, "Which of the following policies or activities would you support (from international organizations or national governments) before using these types of technologies?". Individuals were instructed to "Please choose as many or as few (or none) which you deem to be necessary." As there were seven options available, broken down in terms of "domestic level" and "international level", it was possible that an individual could choose as many as seven options or, conversely, "None of the above". Ordering of the options was randomized within each level. Also, while options were identical for the two types of carbon dioxide removal, there was one difference between options for these groups and those for solar radiation management: with "Global-level market for trading carbon credits and/or offsets" included for carbon removal versus "Creation of new international scientific agency to explore different technologies and start testing and development" for solar radiation management. The decision on which options to include, along with the decision to have one option be different for the two broad technology categories, was based on a review of the literature, specifically the types of governance options discussed[7,11,63–65].

**Climate change beliefs.** The survey asked a set of questions to understand perceptions about climate change plus the potential role of science and technology as a solution. After informing participants, "Now we have a few questions for you about climate change and your personal experience", we asked about their *concern over climate change*: "How worried, if at all, are you about climate change, sometimes referred to as 'global warming'?" Responses were on a five-point scale, with options of: 1 = Not at all worried; 2 = Not too worried; 3 = Somewhat worried; 4=Very worried; 5=Extremely worried; along with a "Don't know" option. We next asked about their *perceived climate harm*, "How much do you think climate change will harm you personally?" Responses were on a four-point scale, with options of: 1 = Not at all; 2 = A little bit; 3 = Somewhat; 4 = A great deal. Next, respondents were asked of their *personal experience with major natural disaster*: "Have you personally experienced the effects of a major natural disaster (e.g., flood, heatwave, wildfire, blizzard) in the last three years?" Possible responses were: Yes; No; Don't know. Finally, participants indicated to what extent they agreed with a statement on *science and technology as a solution to climate change*, "To what extent do you agree with the following statement about science: Science and technology will eventually solve our problems with climate change." Responses were on a balanced five-point scale: 1=Strongly disagree; 2=Tend to disagree; 3=Neither agree nor disagree; 4=Tend to agree; 5=Strongly agree; and a "Don't know" option. These questions have often been used in the literature[13,15,19,33,47,62], with *concern over climate change* and *science and technology as a solution to climate change* adapted from Steentjes et al.[66]. The item on *personal experience with major natural disaster* was a new addition for the survey.

**Information provision**
After initial screening questions, we provided all participants with background information via a short text (and graphic) on carbon dioxide removal and solar radiation management (see Supplementary Fig. 1). This text situated these measures in relation to the negative effects of climate change and, to avoid providing a sense that these solutions were the only ones possible or in any way a kind of panacea, also explicitly referred to mitigation and adaptation as other options. By including all of these in the figure as well, we aimed to further avoid such a priming being imparted. To make sure participants took time to engage with the text, they were required to spend at least 20 seconds on the page before clicking further.

After the background information, individuals were informed they would now be provided with information on a few technologies. To avoid careless reporting, which can particularly be an issue in online surveys[67], we asked that they "please read the texts carefully" as "we want to get your feedback on these technologies". In addition, they were told that their understanding would be checked by one or two short questions and that going faster would not be possible: "In order that you do not feel the need to rush, you will only be able to click to the next slide after 15 seconds have passed."

At this point, individuals were randomly assigned to one of three groups, corresponding to the technology categories: SRM; ecosystem-based CDR; and engineered CDR. We opted for this approach so that participants would have to jointly evaluate multiple technologies. At the same time, to reduce the cognitive load on participants and allow them to engage with the informational materials as much as possible, the groups were composed so that they would be broadly similar to one another, or at least to avoid significant differences. Differences were specifically avoided so that, *inter alia*, perceptions of risks and benefits and overall assessments of the technologies would not be inflated in one direction or another due to their being seen as overly different. For instance, if afforestation and reforestation, DACCS, and space-based geoengineering were grouped together, it was adjudged that having to read and understand information on novel and quite different technologies could (a) be more challenging for participants, thereby affecting engagement and (b) cause individuals to use the most familiar of the technologies, afforestation and reforestation, as an "anchor" when evaluating the others[68]. Participant responses for the others would thereby be overly subject to the context in which this decision was taken. Indeed, other authors[32] have opted to randomly present only one technology to each person to avoid such compositional bias. However, doing so would lead to substantial reductions in the sample size for each of the options and would rule out exploring how individuals evaluate multiple technologies. We view such insights as crucial given the growing discussion of the need for a portfolio of climate-intervention technologies[69] as well as the growing emergence of projects and field trials that employ a combination of options to leverage potential synergies[42]. Similar to other studies[15,33], we opt to present multiple technologies to individuals for consideration, arguing that using clusters of technologies pre-identified from the literature[26,27] allows us to avoid the shortcomings mentioned above. As a result, the three technologies (stratospheric aerosol injection, marine cloud brightening, space-based geoengineering) formed one cluster, and the seven CDR technologies were grouped by more ecosystem-based approaches (afforestation and reforestation, soil carbon sequestration, marine biomass and blue carbon) and rather more engineered approaches (enhanced weathering, biochar, DACCS, BECCS).

Each information text for the climate-intervention technologies followed the same format (see Supplementary Fig. 2), starting with a broad description of how they would (or do) work, followed by 2-3 sentences of more detail and potential benefits, and then 2–3 sentences around prospective risks, and through use of pictures employing the same graphic style. All pictures were designed by the same graphic designer so that they would all have a similar quality and tone. Designs for some of the pictures were iteratively revised, whether to correct any inaccuracies in what was conveyed or so that participants could understand the content more easily. Regarding the information texts, these always began with a sentence about how the measure aimed to "limit the effects of climate change", followed by more detail of how the method worked, while using language as straightforward as possible. Finally, each of the texts concluded with a sentence or two, usually beginning with "However" that mentioned some of the downsides or uncertainties with the technology. Here, we oriented discussions around the negative impacts listed in Fuss et al.[9], Part 2, Table 2, thus with one main socio-economic and one main

environmental risk – the same approach was also tailored to the SRM options. In this way, we could be more systematic in our presentation of potential risks while striving to gather insights on the type of risks which were of greatest concern to participants. At the same time, rather than enumerating the benefits and risks of the technologies, we attempted to provide a description of the techniques. As such, discussions of risks and benefits were intended to be illustrative rather than exhaustive, with the overall aim here to offer participants enough information to feel they could reasonably assess the technologies. One consideration here is that giving information about risks and benefits of climate-intervention technologies, rather than providing a description of the technology, has tended to result in reduced support[14,49]. Such "hint of risk" effects can be common for unfamiliar technologies, and media coverage in general, where a detailed discussion of a subject can elicit a negativity bias[36,70,71]. Information texts were devised to focus on how technologies would (or do) work, rather than provide a detailed account of risks and benefits. Given the novelty of technologies, the determination was also made to avoid "priming" participants via descriptions that overly focused on risks versus benefits, but rather to describe as much as possible how technologies would generally work. Our aim here was to provide a balanced presentation of risks and benefits so that participants could draw their own inferences. One limitation, however, is that despite best efforts to ensure the structure, substance, and format of information texts were consistent for all technologies, unintended differences in complexity may have emerged during the pre-testing and piloting process.

### Statistical analysis and significance testing

Data were analysed using SPSS v28.0. Descriptive statistical analysis included frequency distributions, comparison of group means using nonparametric testing, and hierarchical linear regression analysis (to explore the (sequential) importance of (mean) age and climate change beliefs as distinguishing factors between Global North and Global South). Nonparametric testing was employed given non-normality of the variables. Significance testing related to expressed familiarity across technologies as well as differences in age and climate change beliefs between Global North and Global South employed the independent-samples (two-tailed) Mann-Whitney $U$ test. Differences between group means of technology categories (i.e., SRM, engineered CDR, ecosystem-based CDR) were assessed using pairwise, independent-samples Kruskal-Wallis testing while those between Global North and Global South employed independent-samples (two-tailed) Mann-Whitney $U$ testing. Lastly, significance testing regarding levels of support for research, field trials, and broader deployment employed the related-samples Friedman's two-way analysis of variance by ranks (also confirmed by pairwise related samples Wilcoxon signed-rank testing). In the case of pairwise and multiple comparisons, we employed Bonferroni corrections. Details for the significance testing underlying the results in the main text are provided below for each of the variables considered: familiarity; perceptions of risks and benefits; support for technology; and policy support.

### Reporting summary

Further information on research design is available in the Nature Portfolio Reporting Summary linked to this article.

## Data availability

The datasets generated and/or analyzed in this study, based on the means of the various countries, are publicly available in the Figshare database under accession code: https://doi.org/10.6084/m9.figshare.24893325. Access to the raw data can be made available from the corresponding author on reasonable request, and the totality of the dataset will be made publicly available in full before the conclusion of the GENIE project.

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

## Acknowledgements

This project has received funding from the European Union's Horizon 2020 research and innovation programme under the European Research Council (ERC) Grant Agreement No. 951542-GENIE-ERC-2020-SyG (B.S.), "GeoEngineering and NegatIve Emissions pathways in Europe" (GENIE). The content of this deliverable does not reflect the official opinion of the European Union. Responsibility for the information and views expressed herein lies entirely with the author(s). We are grateful for and acknowledge the assistance of William Lamb and the Mercator Research Institute on Global Commons and Climate Change (MCC) with designing and providing the graphics for each of the technologies as well as for the feedback on survey design from the entire GENIE project team.

## Author contributions

The study was conceived by C.M.B. The detailed design and materials were developed by C.M.B., L.F., S.L., and B.K.S., with C.M.B. and L.F. facilitating the data collection. C.M.B. analyzed the data and wrote the paper with contributions from L.F., S.L., and B.K.S.

## Competing interests

The authors declare no competing interests.

## Inclusion & Ethics Statement

The research has been broadly undertaken with the aim of better understanding public perceptions of climate-intervention technologies, including in the Global South and by means of more qualitative methods that can better elucidate the variability and importance of the local context. At this stage, no local researchers have been included, in part given the global-level focus of the research. As noted in the *Ethical Review Statement*, this research has been approved by the ethics review committee; in addition, the specific roles and responsibilities of those in the author team was discussed prior to the research. Insofar as possible, we have also strived to take into account local and regional research in the citations, though such survey research is rather minimal at the moment.
