## [Peer Review File · Nature Communications]

Public Perceptions and Support of Climate Intervention Technologies across the Global North and Global SouthREVIEWER COMMENTS

Reviewer #1 (Remarks to the Author):

Overall, this is a very interesting paper with a lot of compelling findings. For me, the most significant result is that developing country respondents seem more supportive of SRM than developed country respondents. Personally, this is not surprising – if climate impacts will disproportionately harm the Global South, then using SRM to reduce climate impacts should disproportionately benefit the South – but it clashes with the narratives and claims being made by some of the most prominent critics of SRM, e.g., Biermann et al. 2022. These critics strongly imply that, in making their case against SRM, they are in part channeling the views of people in developing countries who otherwise have little voice in the debate. The results of your research suggest this may not be the case, and that in fact these (Northern) critics are (unsurprisingly) articulating views more prevalent in developed countries and not necessarily shared by people in developing countries. The implication is obvious: that these critics are attempting to seize the moral high ground by falsely attributing their own beliefs to people in poor countries who may not actually share their views, and for whom climate impacts are a more immediate concern.

Of course, yours is a research paper, and you don't need to (and may not wish to) say this. But the significance of this finding deserves more discussion. The reasons why this is so surprising could be elaborated. In a more analytical vein, you could put forward hypotheses about why support for SRM might be stronger in the South than in the North, perhaps drawing on literature that has previously suggested this might be the case.

Related to this is the finding that publics in both the South and the North seem ill-disposed toward bans or moratoria, the approach to SRM favored by the critics noted above. The fact that people not only in developing countries – who appear comparatively favorable toward SRM – but also in developed countries – who are less supportive – seem to dislike the idea of banning the technology also deserves more discussion. Why might this be? Could it be related to concerns about restricting freedom of scientific inquiry? What are the implications of this for proposals for global governance?

Additionally, you note (correctly) that your results support previous findings regarding preferences for measures perceived as more “natural,” which is evident in 1) a general preference for nCDR over eCDR, 2) preferences for more “natural” eCDR options compared to less natural eCDR options, and 3) a general preference for CDR over SRM. However, you neglect to explicitly specify the other end of that spectrum, i.e., technology or engineered systems. This may seem obvious but being explicit about it (including how you identify that pole and why) would make the issue clearer, and potentially make your findings more policy-relevant, for instance in relation to nascent debates in Europe and elsewhere about which types of CDR to support and which might be eligible for inclusion in carbon markets.

On a more editorial level, the text and graphics in section 3.3 need to be cleaned up and clarified along the lines noted below.

I am confused by the letters/numbers scheme you use in many of your tables – I simply don't understand what they mean. What do different letters signify? What do different numbers signify? This needs to be better explained, including by walking through an example (“the value in cell X has a and 1 attached to it. a means Y. 1 means Z. This should be read as ...”).

Assuming the authors address these issues and expand discussions as suggested, I recommend publication.

More specific comments follow.

Line 66—You refer to “the first public-perceptions study examining climate-intervention technologies” – could you identify and cite this?

Line 159—“uneven” should be “disproportionate”.

Table 1—I’m confused by the use of letters a, b, and c. Also, adding rows indicating “benefits” and “risks” to the table would make things clearer.

Line 229—If one assumes that both eCDR and SRM are technology-centric as opposed to nCDR, then one apparent pattern is that approaches perceived as nature-based are consistently viewed as less risky than those based on technology.

Table 2—My comments on Table 1 apply here as well and extend to the use of numbers.

Lines 314-340 (minus Figure 3)—This text seems to refer to composite activity scores across all technologies – is this correct? If so, that needs to be stated clearly, and the method for their derivation needs to be explained. Is there a table or figure that is supposed to accompany it?

Figure 3—This seems to present composite technology scores across all activities – is this correct? If so, that needs to be stated clearly, and the method for their derivation needs to be explained. This figure is not referenced in the text – why is it here?

Table 3—My comments on Table 2 relating to use of letters and numbers apply here as well. Could cells be shaded like Tables 1 and 2? That would aid in understanding.

Line 354—Make clear here that you are shifting the discussion from composite activity scores across all technologies to technology-specific composite activity scores, and that you are equating composite activity scores with “support” and why.

Line 417—Wasn’t “public preferences for activities” already considered in section 3.4?

Table 5a, b—My comments on Table 3 relating to use of letters and numbers and potential shading apply here as well.

Line 464—Statements like “Moreover, the fact that there are no significant differences between technology categories for most of the options that are common to both SRM and CDR (i.e., if the same letters appear in each column for Table 5a or Table 5b) – and seven of eight if we focus only on CDR – is indicative of support for a more explorative than prohibitive policy approach” are intriguing, but difficult to evaluate without a clearer explication of what the letters mean.

Line 518—Your finding about “naturalness” implies that there is less support for “technology” – why not state this explicitly? That would speak directly to ongoing policy issues, for example, the controversy surrounding “engineering-based” methods in relation to the UNFCCC Article 6.4 Supervisory Body.

Lines 550-553—You write “our survey offers evidence that Global South countries are more supportive of SRM technologies than Global North countries, broadly see more benefits to their deployment, and have stronger interest in policies that would facilitate continuing research, development, and oversight.” This seems like by far the most compelling insight generated by your work, and worthy of more discussion.

Methods and Survey Description

Line 22—You write “The only departures from national representativeness were: in Singapore, for

those 55-74" To clarify, you mean that adults ages 55-74 were not well-represented in your sample, correct?

Line 168—Why is "Sunlight reflected out to space" described as an effect of climate change when it is not?

Line 244—Point of clarification – CDR does not limit the effects (impacts) of climate change, but limits climate change itself by reducing (the growth of) the atmospheric stock of CO₂.

Line 248—You write "bioenergy can provide energy for homes and businesses or be stored underground indefinitely" – this is not correct. BECCS could generate power WHILE capturing and storing CO₂ underground.

Reviewer #2 (Remarks to the Author):

A timely study on the perceptions toward climate intervention technologies. Especially novel is the distinction between Global North and South respondents. I support publication as is.

Reviewer #3 (Remarks to the Author):

This study looks at a very large dataset (30 countries, over 30,000 respondents) and thus will be noteworthy and of significance, as most studies have looked at one or a handful of countries. The policy preference section of the paper (3.4) is the most novel contribution and the paper could be framed more in terms of this. (Because of people's low familiarity with SRM and CDR, I am not sure the paper really tells us that much about their preferences, but it does tell us something about respective climate policy preferences between the global North and global South). The basic interpretation of the data – that age and concern about climate change / experience with disasters explain differences in preferences – is largely convincing.

Below are some issues that can be addressed in revision, in rough order of importance:

- Regarding 3.2, it's a bit tricky to ask about "cost-efficient and cheaper than cutting use of fossil fuels" and then say that "This implies that the publics remain somewhat unconvinced about the ability for climate-intervention technologies to function as substitutes for mitigation efforts – even for approaches seen more favorably, such as nature- based CDR." What is this actually intending to measure? Does it mean that people actually believe that the approach is not cheaper and more efficient, or is that they are rejecting the premise of the question, as the interpretation seems to indicate?

The larger issue with this item is that people tend to choose a midpoint when they just don't know. They pick something in the middle when they are not sure what would happen. Most surveys I have seen with low-familiarity climate technologies exhibit this behavior. I would like to see this analyzed beyond just mean values for that reason – more along the lines of Figure 2. It would have been better in my view to have a "not sure / don't know" option that could be analyzed separately. That choice (not to have that option) should be explained and justified. This issue of people picking the middle choice with unfamiliar technologies is the biggest weakness of the paper, but it's one of those things that can't be addressed at this point.

- It would be nice to have a clearer description of how the authors see the relationship between public preferences and decision-making. It is implicit that there is some relationship, and pops up a bit in lines 138-142, but the paper is not explicit about how it works. I am not actually sure about this assertion in lines 148-149 than these are the kind of audiences that researchers and policymakers can expect to encounter when making cases for the use of climate intervention. Probably, the

policymakers will encounter a handful of educated elites who work in think tanks and NGOs and the publics reached through a survey will continue to find this a low salience issue.

- Figure 3 – What’s the rationale for having research, field trials, and deployment collapsed into one figure, when the text above in 3.3 seems to indicate that the differences do matter? This presentation is confusing.

- Abstract: what is “radical”? “Radical” etymologically refers to root; solar geoengineering does not deal with the root cause of climate change. “Radical” can also be seen as subjective / a value judgment. Suggest sticking to more defined terminology. The characterization of radical vs. “self-evident” in the first paragraph of the paper is also not well unpacked; I’m not sure compensation schemes are really self-evident. The paper will be more successful with a wider range of readers if it sticks to straightforwardly descriptive terminology. There is a discussion about how people in the Global South have differing perspectives, and yet these terms may weigh the paper down with value judgements from a particular global North perspective.

In general, the Introduction could be shorter and more direct. Some of the wordier sentences could be cut, and a justification of why the global South and global North are put forward as the two comparative areas could be put forward in a few sentences. This choice in analysis should be explained in brief. Were there differences between individual countries in the GN or GS that were more significant than between the two categories, for example? Were there differences between low income and middle income countries in the GS, or in countries with different educational attainment statistics?

- The brief description of the research design in lines 79-85 is confusing, because it indicates a need to consider portfolios, but that respondents are being asked to choose preferences among different categories of approaches. If portfolios are a given, doesn’t that mean that we are using multiple things at once, and then why is it important whether publics prefer one item out of category A compared to another one?

- For figure 2, it would be great to note in the caption or somewhere that 0 is not at all familiar and 5 is very familiar

- Lines 239-240 – can you really say that people have a preference for nature-based CDR vs. engineered CDR and SRM if respondents were not asked to compare them?

- Lines 547-548 – “there are signals in the Global South of a blurring across the technology categories in a way that is not present in the Global North” – that could be unpacked more.

Reviewer comments	Response from the authors
Reviewer #1: Overall, this is a very interesting paper with a lot of compelling findings. For me, the most significant result is that developing country respondents seem more supportive of SRM than developed country respondents. Personally, this is not surprising – if climate impacts will disproportionately harm the Global South, then using SRM to reduce climate impacts should disproportionately benefit the South – but it clashes with the narratives and claims being made by some of the most prominent critics of SRM, e.g., Biermann et al. 2022. These critics strongly imply that, in making their case against SRM, they are in part channeling the views of people in developing countries who otherwise have little voice in the debate. The results of your research suggest this may not be the case, and that in fact these (Northern) critics are (unsurprisingly) articulating views more prevalent in developed countries and not necessarily shared by people in developing countries. The implication is obvious: that these critics are attempting to seize the moral high ground by falsely attributing their own beliefs to people in poor countries who may not actually share their views, and for whom climate impacts are a more immediate concern.	Thank you, kindly, for your thoughtful review. On this particular point, we concur that this result is one of the key findings – and that it tends to contradict some of the contentions in the literature. In fact, we have noted as much in the final paragraph of the Results section and at a few points in the Discussion and Conclusion. We have also added some language to the latter section to this effect to make our argument stronger and punchier on this point – while of course acknowledging that more research is needed here to better foreground the multiplicity of viewpoints and perspectives which could be present in the Global South (and Global North, for that matter).
Of course, yours is a research paper, and you don't need to (and may not wish to) say this. But the significance of this finding deserves more discussion. The reasons why this is so surprising could be elaborated. In a more analytical vein, you could put forward hypotheses about why support for SRM might be stronger in the South than in the North, perhaps drawing on literature that has previously suggested this might be the case.	Thank you for your encouragement on this point. We agree in part, and have added the new paragraph at the end of the Conclusion to buttress the argument outlined above. Regarding some particular reasons for why support may be stronger in the Global South than North, this is what is intended by Table 4 in Section 3.3 and the related discussion – notably, that perceptions and experiences of climate harm as well as beliefs in science and technology as a solution to climate change are all significantly higher across the Global South cohort. There is also the matter of the age of the nationally representative samples

	(itself illustrative of the mean/median ages of the national populations). Age is significantly predictive of technology support, particularly for SRM: i.e., from page 16, “there are initial indications that it is not necessarily whether a country is in the Global North or Global South but rather the youthfulness of the population that broadly corresponds to the level of support for climate-intervention technologies.” In total, these findings identify prominent areas for future exploration, and which can serve as the foundation for further empirical research.
Related to this is the finding that publics in both the South and the North seem ill-disposed toward bans or moratoria, the approach to SRM favored by the critics noted above. The fact that people not only in developing countries – who appear comparatively favorable toward SRM – but also in developed countries – who are less supportive – seem to dislike the idea of banning the technology also deserves more discussion. Why might this be? Could it be related to concerns about restricting freedom of scientific inquiry? What are the implications of this for proposals for global governance?	This raises an important issue – and one that is certainly at odds with some strands of the discourse. It is for this reason that we write (on page 19) the following: “There are very few points of consensus between Global North and Global South; it is however illustrative that the two are broadly agreed on their support (or lack thereof) for an international ban or moratorium (row 4). Indeed, support for this policy option is lacking for all technology types and irrespective of cohort, indicating the limited resonance for the public of proposals of the kind debated in academic and policy circles (Biermann et al. 2022; McLaren and Corry 2021).” This point is then picked up again in the Discussion (page 21): “There is comparatively less interest at the moment – at least when presented in a general manner – in the kinds of bans or moratoria, i.e., on SRM, discussed in the literature (Biermann et al. 2022), whether at national or international level...”. As to why this is the case, we are reluctant to speculate – as the Reviewer rightly noted above, this is empirical research after all. It also seems to us more than sufficient to let

	such clear results speak for themselves (with all apparent caveats befitting such research duly noted). It may be for the reasons proposed by the Reviewer, or simply since the understanding of risks (e.g., of SRM) do not at present rise to the level for such measures to be taken.
Additionally, you note (correctly) that your results support previous findings regarding preferences for measures perceived as more “natural,” which is evident in 1) a general preference for nCDR over eCDR, 2) preferences for more “natural” eCDR options compared to less natural eCDR options, and 3) a general preference for CDR over SRM. However, you neglect to explicitly specify the other end of that spectrum, i.e., technology or engineered systems. This may seem obvious but being explicit about it (including how you identify that pole and why) would make the issue clearer, and potentially make your findings more policy-relevant, for instance in relation to nascent debates in Europe and elsewhere about which types of CDR to support and which might be eligible for inclusion in carbon markets.	We agree to an extent (and we note that we also touch on this point below in relation to a similar comment). While we highlight the broad preference for more “natural” options, doing so also to emphasize how this relates to findings in the extant literature, it seems to us a step too far to extrapolate say to say that individuals are broadly dismissive of more “technological” options. Indeed, we are skeptical whether such a strong takeaway can be made regarding the role of “technology” given that we (deliberately) employed no such framing in the study as well as since, according to the level of agreement with the statement about “science and technology as solution to climate change”, participants tended to be generally positive about the role of science and technology in this context – albeit more so in the Global South. Caveats aside, pursuant to your comment below about line 229, we have broadly adopted your implication and language here to make this point more explicitly, as we feel that this helps to enrich the discussion. Thanks! Lastly, regarding a general preference of CDR vis-à-vis SRM, we wish to clarify that we also emphasize that “there are notable exceptions” here (page 20). More than anything, the findings underscore that it is neither desirable nor illustrative to speak of CDR as a broad category. We have added language to further clarify this point in the Discussion. Motivated by your comment, we have revised the fourth paragraph in the Discussion and

	Conclusion section, so that it now reads as follows: “Overall, the divergence in perceptions across CDR technologies highlights the need for caution when speaking of CDR as a broadly coherent category. Whereas many studies have examined perceptions of “CDR in general” (or “SRM in general”), our findings underscore the value of a granular approach that examines perceptions on the technology level (as done by Carlisle et al. 2020; Cox et al. 2020; Sweet et al. 2021; Wenger et al. 2021; Bellamy 2022).”
On a more editorial level, the text and graphics in section 3.3 need to be cleaned up and clarified along the lines noted below. I am confused by the letters/numbers scheme you use in many of your tables – I simply don’t understand what they mean. What do different letters signify? What do different numbers signify? This needs to be better explained, including by walking through an example (“the value in cell X has a and 1 attached to it. a means Y. 1 means Z. This should be read as ...”). Assuming the authors address these issues and expand discussions as suggested, I recommend publication.	Thanks – we have now gone through the entire text and made revisions to improve clarity on how the letters and numbers should be interpreted, including through the addition of clarifying text related to each Table. We appreciate the encouragement to do so.
More specific comments follow. Line 66—You refer to “the first public-perceptions study examining climate-intervention technologies” – could you identify and cite this?	Good point – thanks! The study in question is now cited and included in the references: Mercer, A. M., Keith, D. W., & Sharp, J. D. (2011). Public understanding of solar radiation management. Environmental Research Letters, 6(4), 044006.
Line 159—“uneven” should be “disproportionate”.	Agreed, changed.
Table 1—I’m confused by the use of letters a, b, and c. Also, adding rows indicating “benefits” and “risks” to the table would make things clearer.	Thank you for highlighting these points – rows specifying which items relate to “Benefits” and “Risks” have now been added.

	We agree that such an important point should be made more clearly. We have revised this section along with specifically adding the following: “Looking at the rows related to the benefits and risks in Table 1, if distinct letters accompany the numbers for the categories, perceptions are significantly different (according to nonparametric testing).” We have also undertaken significant revisions to the explanatory note below Table 1: “If different letters are present for columns of a given row (e.g., “a” versus “b”), this indicates that significant differences exist among perceived benefits/risks of the respective technology categories ($p < 0.05$), according to pairwise, independent-samples (two-tailed) Kruskal-Wallis H testing, with adjustments for multiple comparisons using the stepwise step-down method (for technologies). If the letters are the same (i.e., “a” appears throughout), no significant differences exist.”
Line 229—If one assumes that both eCDR and SRM are technology-centric as opposed to nCDR, then one apparent pattern is that approaches perceived as nature-based are consistently viewed as less risky than those based on technology.	Agreed, we now say as much explicitly (on page 9): “As such, if one assumes engineered CDR and SRM are generally more technology-centric, one potential implication is that approaches perceived to be more natural or grounded in ecosystems tend to be viewed as less risky than those relying more on (novel) technologies.”
Table 2—My comments on Table 1 apply here as well and extend to the use of numbers.	As before, the recommended rows for “Risks” and “Benefits” have been added. Also, the following sentence has been added (on page 9): “Comparing the columns of technology categories for “Global North” and “Global South” in Table 2, if the numbers differ (i.e.,

	if there is a “2” for Global South), perceptions of the cohorts are significantly different.” This accompanies the explanatory sentences included in the Note below Table 2.
Lines 314-340 (minus Figure 3)—This text seems to refer to composite activity scores across all technologies – is this correct? If so, that needs to be stated clearly, and the method for their derivation needs to be explained. Is there a table or figure that is supposed to accompany it?	Apologies for any confusion here – the level of a support for a given activity is only contrasted at a technology level. Notably, the first sentence in Section 3.3. reads: “We queried participants on their support for three kinds of activity regarding each technology, roughly corresponding to stage and scale.” When we use “mean” here, we refer to the mean across the entire sample population for a given technology – not as a composite across all technologies. We do concur, however, that this is muddled by presenting the means for each kind of activity across all technologies. This was necessitated by it not being feasible to provide the values for each activity at a technology level. We have substantially rewritten this paragraph in order to better distinguish between these two aspects, e.g., while including the following in particular: “Though nonparametric testing considered support for a given activity for each of the technologies separately, the broad nature of support by activity can be gleaned by looking at composite means for all technologies (standard deviation in parentheses): 3.893 (0.400) for small-scale field trials vs. 3.733 (0.478) for research or 3.699 (0.488) for broader deployment.”
Figure 3—This seems to present composite technology scores across all activities – is this correct? If so, that needs to be stated clearly, and the method for their derivation needs to be explained. This figure is not referenced in the text – why is it here?	This is correct – these are the composite scores across all technologies. We strived to convey this through the title of Figure 3 and the heading on the Y-axis. We agree on the need to avoid confusion and, as such, we now include the following

	information in the Note below Figure 3 (mirroring that included for Table 3): “Responses reflect overall support for technology, across activities for research, small-scale field trials, and broad deployment. The items were combined by taking their average, given the high levels of correlation between them (i.e., the lowest Spearman’s rho correlation was 0.956 between small-scale field trials and broader deployment). Scale is from 1 to 5: 1=“Strictly reject” 2= “Somewhat reject”, 3= “Neither reject nor support” 4= “Somewhat support”, 5=“Fully support”, and “Don’t know”; the latter were characterized as missing values for the sake of analysis.”
Table 3—My comments on Table 2 relating to use of letters and numbers apply here as well. Could cells be shaded like Tables 1 and 2? That would aid in understanding.	We have now added the following paragraph preceding the results for Table 3: “To understand public preferences for the technology options, we also examined how the level of support (composite across all activities) differed between the technologies – and in relation to the technology categories. Looking at the columns in Table 3, if the letter accompanying the respective technologies is the same – e.g., the three SRM options and enhanced weathering in the “Global” column are accompanied by “a” – level of support was not found to significantly differ (according to nonparametric testing). Also, since level of support significantly increased with each new letter, an overall ranking of the technology options is thereby attained. Similar to Table 2, if the numbers for a row (technology) are different for the Global North and Global South columns, the level of support was significantly different between the two cohorts.” This is accompanied by significant revisions to the explanatory Note below Table 2. Regarding shading, we have discussed this option in detail (even designing a potential

	Table). Ultimately, we have determined that this is likely to hinder rather than aid understanding. The main reason is that the types of comparisons being made differ between the various Tables. For one, the shading in Tables 1 and 2 relates to the “1-7” scale employed for the variable – which is illustrative given that there is a wider dispersion, both across the technology categories and around the mid-point of the scale. Moreover, unlike Tables 1 and 2, Table 3 does not consider significant differences for each row in isolation but across rows for a given column (using the stepwise step-down method). For instance, the fact that the three SRM options (and enhanced weathering) are denoted by an “a” for the “Global” column entails that the level of support for these options does not significantly differ. A further complication is that no shading scheme could at the same time also reflect significant differences between Global North and Global South. This issue is somewhat mitigated by the fact that perceptions of the two cohorts are found to differ for every technology except afforestation and reforestation. In any case, this again highlights the key differences between the Tables.
Line 354—Make clear here that you are shifting the discussion from composite activity scores across all technologies to technology-specific composite activity scores, and that you are equating composite activity scores with “support” and why.	Thanks – having clarified above the (limited) extent to which we discuss composite means for activities, any concerns about such a shift should broadly be addressed. We have however added the following as an introductory statement to this paragraph to further avoid confusion: “We also explored (at a technology level) if the pattern of support for the three kinds of activities differed between the Global North and Global South cohorts.”

	As far as the conceptualization of “support”, we kindly draw the Reviewer’s attention to the Notes below Figure 3 and Table 3, i.e.: “Responses reflect overall support for technology, across activities for research, small-scale field trials, and broad deployment. The items were combined by taking their average, given the high levels of correlation between them (i.e., the lowest Spearman’s rho correlation was 0.956 between small-scale field trials and broader deployment).” Given word limits and in order to otherwise hamper readability, we decide against inserting such a statement directly into the main text.
Line 417—Wasn’t “public preferences for activities” already considered in section 3.4?	This is correct – however the language here, mirroring the question that was presented to the participants (“Which of the following policies or activities would you support (from international organizations or national governments) before using these types of technologies?”), is understood more in the context of policy. This wording is employed given that some of the options were not policies per se, such as information and engagement campaigns to consult and inform public. To avoid confusion, we have now slightly reworded to “public preferences related to policies or governance activities”.
Table 5a, b—My comments on Table 3 relating to use of letters and numbers and potential shading apply here as well.	Similar to above, we have added the following at the beginning of the first of the results paragraphs related to the Tables: “Comparing Tables 5a and 5b, notably if the numbers of the relevant columns differed between the two tables...”. In addition, we have undertaken significant revisions to the Note below Tables 5 and 5b (as well as for Table 4). Regarding the letters (as the Reviewer notes in below), the text includes the following by

	way of clarification: “(i.e., if letters in the two columns are identical)”. A similar statement also appears in the following paragraph: “(i.e., the letters are the same in columns 2 and 3 for both Tables 5a and 5b)”. Similar to Table 3 (see above), we determined that shading would not be as illustrative here as for Tables 1 and 2.
Line 464—Statements like “Moreover, the fact that there are no significant differences between technology categories for most of the options that are common to both SRM and CDR (i.e., if the same letters appear in each column for Table 5a or Table 5b) – and seven of eight if we focus only on CDR – is indicative of support for a more explorative than prohibitive policy approach” are intriguing, but difficult to evaluate without a clearer explication of what the letters mean.	Thanks – with the aforementioned changes and explication throughout, we feel that this problem is greatly reduced. For good measure, we have also revised this part to hopefully be clearer: “Moreover, the fact that there are no significant differences (according to nonparametric testing) between SRM and CDR for most of the policy options (i.e., if letters in the two columns are identical) is broadly indicative of support for a more explorative than prohibitive policy approach. Given overall support for such institutions, notably in the Global South, such an approach could potentially be pursued under the auspices of an international architecture able to conduct oversight and offer assessment.”
Line 518—Your finding about “naturalness” implies that there is less support for “technology” – why not state this explicitly? That would speak directly to ongoing policy issues, for example, the controversy surrounding “engineering-based” methods in relation to the UNFCCC Article 6.4 Supervisory Body.	We take your point, and it is certainly one that merits further discussion and debate. We are skeptical however whether such a strong takeaway can be made regarding the role of “technology” based on the present results. After all, we (quite deliberately) avoided any framing in terms of nature or technology – any references throughout the text to “nature” are employed given how such technologies tend to be described and named in the literature. In a sense, it could also be argued that what is most determinative here is the “proximity” to ecosystems – though this can also have a

	pejorative impact, as ostensibly is the case for enhanced weathering. Moreover, according to the level of agreement with the statement about “science and technology as solution to climate change”, we find that the participants tended to be generally positive about the role of science and technology in this context – albeit more so in the Global South. It is for these reasons, inter alia, that we find it necessary to not be more explicit here. As is for instance demonstrated among the different examples of CDR, public perceptions can differ markedly depending on the exact approach employed – and there is a need for further research into how the public might evaluate different permutations of how even a given approach is employed.
Lines 550-553—You write “our survey offers evidence that Global South countries are more supportive of SRM technologies than Global North countries, broadly see more benefits to their deployment, and have stronger interest in policies that would facilitate continuing research, development, and oversight.” This seems like by far the most compelling insight generated by your work, and worthy of more discussion.	Thank you, we agree, and have expanded upon this with new text at the end of the Conclusion.
Methods and Survey Description Line 22—You write “The only departures from national representativeness were: in Singapore, for those 55-74” To clarify, you mean that adults ages 55-74 were not well-represented in your sample, correct?	First of all, we kindly thank you for your close and attentive reading of the “Methods and Survey Description”. Your comments here are greatly appreciated and highly useful. To clarify, it is not that they were not “well-represented” but rather not at levels reflecting their national share. We have added the following sentence to hopefully make this clearer: “In such cases, these groups were still broadly represented, albeit at levels falling short of being national representative.”
Line 168—Why is “Sunlight reflected out to space” described as an effect of climate change when it is not?	The phrase that the Reviewer is highlighting falls within a smaller box with the heading “Effects on Earth’s climate”. In this regard, these two effects should be understood as

	describing the “mediating” factors by which the other factors influence climate change. These are themselves not “effects of climate change” but rather influence the “effects on the Earth’s climate” – they are also crucial to give an initial sense, in brief form, of why measures such as SRM and CDR are being examined.
Line 244—Point of clarification – CDR does not limit the effects (impacts) of climate change, but limits climate change itself by reducing (the growth of) the atmospheric stock of CO2.	We take your point well – though it might fail to capture some of the nuance involved, it was important for the information texts to, as much as possible, follow the same structure to improve readability and avoid the introduction of any unintended biases. For this reason, all of the texts begin with “This aims to limit the effects of climate change” (also mirroring the discussion of the introductory text), followed by a short description of how this would be done.
Line 248—You write “bioenergy can provide energy for homes and businesses or be stored underground indefinitely” – this is not correct. BECCS could generate power WHILE capturing and storing CO2 underground.	Thank you for this clarification – we would humbly note, however, that while the two can be done in tandem to a certain extent, there is an unavoidable tradeoff given that the use of CCS with bioenergy involves an “energy penalty” on power plants. There is substantial disagreement on the size of this penalty in the literature (e.g., Gustafsson et al. 2021). For the purposes of the current study, the crucial aspect was to avoid imparting the sense that the two were perfect complements – as this might (incorrectly) prejudice perceptions. Reference: Gustafsson, K., Sadegh-Vaziri, R., Grönkvist, S., Levihn, F., & Sundberg, C. (2021). BECCS with combined heat and power: Assessing the energy penalty. International Journal of Greenhouse Gas Control, 110, 103434.
Reviewer #2: A timely study on the perceptions toward climate intervention technologies. Especially novel is the	Thank you very, very much – in order to be meriting of such a glowing review, we have endeavored to improve the manuscript further based on the other Reviewer comments.

distinction between Global North and South respondents. I support publication as is.	
Reviewer #3: This study looks at a very large dataset (30 countries, over 30,000 respondents) and thus will be noteworthy and of significance, as most studies have looked at one or a handful of countries. The policy preference section of the paper (3.4) is the most novel contribution and the paper could be framed more in terms of this. (Because of people’s low familiarity with SRM and CDR, I am not sure the paper really tells us that much about their preferences, but it does tell us something about respective climate policy preferences between the global North and global South). The basic interpretation of the data – that age and concern about climate change / experience with disasters explain differences in preferences – is largely convincing.	Thanks - we appreciate your accurate and fair reading of the manuscript. Your sound suggestions for improvement have been largely adopted as well. We also wholeheartedly agree on your centering of the policy preference results. We have revised the paper with an eye towards highlighting these more strongly, including as part of a new paragraph at the end of the paper.
Below are some issues that can be addressed in revision, in rough order of importance: - Regarding 3.2, it’s a bit tricky to ask about “cost-efficient and cheaper than cutting use of fossil fuels” and then say that “This implies that the publics remain somewhat unconvinced about the ability for climate-intervention technologies to function as substitutes for mitigation efforts – even for approaches seen more favorably, such as nature- based CDR.” What is this actually intending to measure? Does it mean that people actually believe that the approach is not cheaper and more efficient, or is that they are rejecting the premise of the question, as the interpretation seems to indicate?	This item is intended as a form of “moral hazard” measure that considers the economic rationale of the relevant technology vis-à-vis climate mitigation / reducing fossil emissions. To the extent that participants are in agreement with this item – we highlight that agreement is consistently strongest for nature-based CDR and lowest for SRM – the expectation is that the particular technology category could be evaluated as a potential substitute for climate mitigation. Given that the results signal possible concerns over SRM being viewed in such a manner, particularly in the Global North, we find limited evidence that they are rejecting the premise of the question. We have however revised this sentence to the following to hopefully clear up any potential confusion: “This implies that the publics remain less willing to countenance climate-intervention technologies when presented in this manner – that is, as substitutes for climate mitigation.

	Though this is even true for approaches seen more favorably, such as nature-based CDR, there is significantly lower agreement for SRM in this regard.”
The larger issue with this item is that people tend to choose a midpoint when they just don’t know. They pick something in the middle when they are not sure what would happen. Most surveys I have seen with low-familiarity climate technologies exhibit this behavior. I would like to see this analyzed beyond just mean values for that reason – more along the lines of Figure 2. It would have been better in my view to have a “not sure / don’t know” option that could be analyzed separately. That choice (not to have that option) should be explained and justified. This issue of people picking the middle choice with unfamiliar technologies is the biggest weakness of the paper, but it’s one of those things that can’t be addressed at this point.	Yes, thank you for raising this important point. The tendency to “straightline” responses in surveys and/or default to the mid-point is a consistent risk, particularly for low-familiarity technologies (as the Reviewer notes). Any advantages from use of an opt-out option, such as “Don’t know”, need to be carefully weighed against, e.g., the added insights to be gained from nudging survey participants to take a stance one way or the other. This is particularly true in the case of unfamiliar technologies. Often, when opt-out options are used too liberally, this can encourage participants to always take this answer. This is one reason why, while we do have “Don’t know” as an option elsewhere in the survey (e.g., for the question about levels of support), we decided against doing so for the risk-benefit items. Also, the presumption that participants have simply defaulted to the mid-point is undercut by, inter alia, (i) the significant differences across technology categories, (ii) the pattern of such differences to be more pronounced for the benefit vis-à-vis risk items, and (iii) for the particular item in question, the fact that (driven by results in the Global North) the value for SRM is one of the most extreme of any in Tables 1 and 2. If anything, the fact that some participants might have resorted to the mid-point would then imply that the differences here could actually underestimate the differences that do exist. One final point (as we discussed in Section 3.1, footnote 4): response styles vary across

	countries and cultures. The fact that there is a greater proclivity towards “extreme” response styles in Western countries is countered by use of more mid-point value in Asian ones. As such, it cannot be assumed that there is a general default to the mid-point. That being said, we do fundamentally agree that discussion of this point could be useful for interested readers. We have therefore undertaken revisions to this effect in the “Methods and Survey Description”, specifically for the section on “Perceptions of risks and benefits” under “Survey measures”: “We opted against the inclusion of “Don’t know” as an answer option for these items given the potential insights to be gained from nudging survey participants to take a stance one way or the other. In contrast, over-use of such an approach can encourage participants to always take this answer. Though we do include “Don’t know” as an option elsewhere in the survey, we deemed it useful and informative to not afford individuals this choice for risk-benefit items.”
- It would be nice to have a clearer description of how the authors see the relationship between public preferences and decision-making. It is implicit that there is some relationship, and pops up a bit in lines 138-142, but the paper is not explicit about how it works. I am not actually sure about this assertion in lines 148-149 than these are the kind of audiences that researchers and policymakers can expect to encounter when making cases for the use of climate intervention. Probably, the policymakers will encounter a handful of educated elites who work in think tanks and NGOs and the publics reached through a survey will continue to find this a low salience issue.	Thanks for highlighting this point – after some discussion, we agree that the assertion in lines 148-149 is somewhat outside the scope of this paper / better addressed through another paper that specifically focuses on this. We have therefore addressed your comment in a few ways. First, we deleted the second part of this statement. Second, we revised and extended the first part to better highlight the key results of the current paper (and in line with your comments about the kinds of actors currently driving the discourse around climate-intervention technologies): “Nonetheless, we contend that a survey with the current scope, scale, and inclusiveness of the Global South provides crucial insights into where public perceptions and familiarity with these technologies is at present.

	Crucially, it also underscores how perceptions of publics around the world differ, to one another and in relation to experts.” Third and finally, we pick up and expound on the relevance of our results as a corrective to the broadly expert-led discourse in the final paragraph of the Discussion and Conclusion section.
- Figure 3 – What’s the rationale for having research, field trials, and deployment collapsed into one figure, when the text above in 3.3 seems to indicate that the differences do matter? This presentation is confusing.	It is true that there are significant differences between perceptions of the three activities. At the same time, the pattern of how the perceptions of the technologies related to one another is largely consistent, irrespective of the kind of activity. For this reason, we decided it would be repetitive to have figures for each of the activities. Moreover, as previously noted in the Note below Table 3 (and in the Methods and Survey Description section), there was a high level of correlation across the three measures for support. We have therefore included a similar statement now below Figure 3 as well: “Responses reflect overall support for technology, across activities for research, small-scale field trials, and broad deployment. The items were combined by taking their average, given the high levels of correlation between them (i.e., the lowest Spearman’s rho correlation was 0.956 between small-scale field trials and broader deployment). Scale is from 1 to 5: 1=“Strictly reject” 2= “Somewhat reject”, 3= “Neither reject nor support” 4= “Somewhat support”, 5=“Fully support”, and “Don’t know”; the latter were characterized as missing values for the sake of analysis.”
- Abstract: what is “radical”? “Radical” etymologically refers to root; solar geoengineering does not deal with the root cause of climate change. “Radical” can also	To be clear, we did not characterize climate-intervention technologies as a whole as “radical” but rather stated that these were “often radical” – even in one place putting

be seen as subjective / a value judgment. Suggest sticking to more defined terminology. The characterization of radical vs. “self-evident” in the first paragraph of the paper is also not well unpacked; I’m not sure compensation schemes are really self-evident. The paper will be more successful with a wider range of readers if it sticks to straightforwardly descriptive terminology. There is a discussion about how people in the Global South have differing perspectives, and yet these terms may weigh the paper down with value judgements from a particular global North perspective.

this in quotes in order to convey that we were gesturing towards one particular and commonplace way that this term is used.

Namely, there is a notable segment of the public and civil society which frequently employ such language. Here is a sampling of articles, some of which quite recent, which have specifically discussed climate-intervention technologies in such terms:

- McKie, R. (2023, July 30). Radical ways to fix the Earth: Are they magic bullets or just band-aids? The Observer.
<https://www.theguardian.com/environment/2023/jul/30/radical-ways-to-fix-the-earth-are-they-magic-bullets-or-just-band-aids>
- Flavelle, C. (2020, October 28). As Climate Disasters Pile Up, a Radical Proposal Gains Traction. The New York Times.
<https://www.nytimes.com/2020/10/28/climate/climate-change-geoengineering.html>
- Jones, B. (2017, November 16). Who’s right about this radical plan to cool the planet? NBC News.
<https://www.nbcnews.com/mach/science/who-s-right-about-radical-plan-cool-planet-ncna821456>
- Levey, M. (2022, June 10). Solar Geoengineering Would Be Radical. It Might Also Be Necessary. Freakonomics. Retrieved October 5, 2023, from
<https://freakonomics.com/podcast/solar-geoengineering-would-be-radical-it-might-also-be-necessary/>
- Groch, S. (2020, February 13). Space mirrors, fake volcanoes: The radical plans to fix the climate. The Sydney Morning Herald.
<https://www.smh.com.au/national/space-mirrors-fake-volcanoes-the-radical->

	plans-to-fix-the-climate-20200122-p53tq3.html We appreciate the more rigorous and etymologically correct application of this term which the Reviewer references. Indeed, the one place in the main text where we use the term “radical”, we make sure to reference a recent paper by Morrison et al. (2022) which is similarly nuanced. Given our engagement with public perceptions around these technologies and the consistent use of such language in relation to them, we determined that it was important to make (limited and appropriately cautious) use of it as well. We have however now deleted one of the references to “radical” (in the Abstract) while changing the relevant adverb to “potentially” rather than “often”, to better convey some uncertainty here.
In general, the Introduction could be shorter and more direct. Some of the wordier sentences could be cut, and a justification of why the global South and global North are put forward as the two comparative areas could be put forward in a few sentences.	Fair point, and one we have implemented in full, shortening the introduction by about 25% in length.
This choice in analysis should be explained in brief. Were there differences between individual countries in the GN or GS that were more significant than between the two categories, for example? Were there differences between low income and middle income countries in the GS, or in countries with different educational attainment statistics?	The Reviewer raises an important point – and of course there are other ways to characterize the different countries. For the present paper, in view of the cited research and emerging discussions around beliefs and perceptions of publics in the Global South regarding climate-intervention technologies, this was determined to be the most urgent vein for exploration. As far as differences between countries within the respective cohorts, there are of course exceptions regarding how a public in a particular country views a technology (more positively or negatively. For reasons to be explored further, Italian participants were

	much more negative about enhanced weathering as an example. Though Turkey was grouped as Global North in keeping with the classification which we employed (and as a candidate country for accession to the European Union), the responses of participants from this country were more indicative of a country in the Global South – we discussed this in relation to the familiarity item at the end of Section 3.1. We might also say that many western and northern European countries were less positive and supportive than, for instance, “Anglosphere” countries – though mostly for SRM options and/or some of the engineered CDR ones. On the whole, however, such differences were certainly not “more significant than between the two categories”. Together with the prominence of the Global North / Global South in the literature and media discourse, this explains why we focus our analysis in this way. To make this clearer, we have added the following to the Abstract: “We focus on this distinction given emergent debates, and lack of research, regarding perspectives of those in the Global South”; along with a broadly similar statement at the end of the Introduction: “These cohorts are employed given the questions (and a lack of research) around public perceptions in the Global South of climate-intervention technologies, along with the prominence of this distinction in the literature.”
- The brief description of the research design in lines 79-85 is confusing, because it indicates a need to consider portfolios, but that respondents are being asked to choose preferences among different categories of approaches. If portfolios are a given, doesn't that mean that we are using multiple things at once, and then why is it important whether publics prefer one item out of category A compared to another one?	We agree and have removed it in our efforts to trim the Introduction, in line with your comment above.

- For figure 2, it would be great to note in the caption or somewhere that 0 is not at all familiar and 5 is very familiar	Done.
- Lines 239-240 – can you really say that people have a preference for nature-based CDR vs. engineered CDR and SRM if respondents were not asked to compare them?	We take your point and appreciate your raising it for consideration – it is true that it is not the same participants who were asked to evaluate all technologies. That would of course been overly demanding, not least given that our survey is already one of only very few to have people evaluate more than one option. We would however clarify that we nowhere mention “people” or the participants having this preference but rather that, for the publics / the sample population as a whole, such a preference is apparent. Given that we have employed random assignment to these categories, we contend that such a statement can be defended at the population level. We have re-checked the document to ensure that everywhere “preference” appears, it is used in the way mentioned above. We are now doubly confident that this is the case.
- Lines 547-548 – “there are signals in the Global South of a blurring across the technology categories in a way that is not present in the Global North” – that could be unpacked more.	After much discussion, we have decided to delete this point. We agree that it requires more unpacking and, while there are signals in the data, the amount of evidence supporting such a statement is probably too limited to justify such a discussion at the present moment.

REVIEWER COMMENTS

Reviewer #1 (Remarks to the Author):

I thank the authors for expanding on substantive discussions along the lines suggested. But I remain at a loss when it comes to your tables. While I appreciate your attempt to revise and clarify them, I still find these tables very challenging to navigate and understand, particularly your varied use of letters and numbers to denote an array of different relationships. It may be that you're trying to pack too much information into tables, or that a tabular format isn't the best way to present some of your data. I would think that using colors and shading and/or patterns like hatching and stippling would be a big help, but you note that you've tried that and it doesn't work. Perhaps disaggregating your tables, possibly using Supplementary Materials, would improve things.

Take Tables 3-5 for example:

- In Table 3, "a,b,c" seem to signify different levels of support across technologies, in Table 4 "a,b,c" seem to signify different levels of support across cohorts, while in Table 5 "a,b,c" seem to signify different levels of support across policies.
- In Table 3, "a,b,c" seem to be relevant only within columns and not across columns (even though the same letter may be used in multiple columns), whereas in Tables 4 and 5 "a,b,c" seem to be relevant only across columns and not within columns (even though in Table 5 the same letter may be used in multiple rows).
- In Table 3, "a,b,c" seem to signify increasing levels of support, but it is not clear if that is the case in Tables 4 and 5.
- In Table 3, "1,2" seem to signify different levels of support across cohorts within Table 3, whereas in Table 5 "1,2" seem to signify different levels of support across cohorts in Tables 5a and 5b.

The other tables have similar issues of inconsistent signification and orientation and imprecision leading to confusion. I think you need to alter your approach to data presentation in section 3, especially the tables. It may be helpful to bring in an additional coauthor with experience in communicating complex data in understandable ways. At present, I fear that many readers will find these tables and associated results confounding at best, and incomprehensible at worst. Unless these are revamped, I cannot support publication.

My remaining comments are not dealbreakers, but it would be nice if they were addressed where possible, particularly the final two.

You respond to my call for discussing greater support for SRM in the Global South than the Global North more extensively by pointing to Table 4 and the related discussion. Fair enough but note that the reason discussion of SRM is not always apparent is because you have chosen to lump it together with CDR as "climate intervention technologies," which tends to obscure some of the analysis. For example, in practice, discussions about a moratorium or ban focus almost exclusively on SRM, not CDR, yet you discuss it broadly in terms of "technology"—by lumping SRM and CDR together, the potential policy relevance of findings related to proposed bans on SRM are less clear to the reader. Your statement that "the findings underscore that it is neither desirable nor illustrative to speak of CDR as a broad category" applies even more strongly to "climate intervention technologies" as a broad category.

It's still not clear whether Figure 3 is intended to accompany the paragraph immediately preceding it. If it is, that is confusing since the text refers to "composite means for all technologies" but the figure is broken down by individual technology.

Regarding your response to my comment about BECCS, I think you miss the point—as written, your statement "bioenergy can provide energy for homes and businesses or be stored underground indefinitely" suggests bioenergy can be stored underground indefinitely, but what you mean is carbon. Further, if your statement is revised to read "bioenergy can provide energy for homes and businesses

or carbon can be stored underground indefinitely," this is still not correct as BECCS is envisioned as doing both (with an energy penalty), not one or the other.

Reviewer #3 (Remarks to the Author):

I thought this revision did a good job addressing the concerns raised by the reviewers, and that this version would be good to publish.

I am going to admit to not being totally won over by the explanatory notes in the tables - they still leave me a bit confused - but will go with it. If it seems to be too wordy / too much info, maybe a simplified version could be presented here with a more detailed version in the SI.

I appreciate the new final paragraph, but have to admit that the last sentence kind of opens up a whole new avenue — what does "direct engagement with representative publics in the Global South" entail and how can it be done? It's the kind of statement that makes you think a whole new chapter needs to be written and that more is coming, rather than the kind of statement that neatly wraps a paper. Consider maybe rewording the last clause or adding one more sentence at the end.

Reviewer comments	Response from the authors
Reviewer #1: I thank the authors for expanding on substantive discussions along the lines suggested. But I remain at a loss when it comes to your tables. While I appreciate your attempt to revise and clarify them, I still find these tables very challenging to navigate and understand, particularly your varied use of letters and numbers to denote an array of different relationships. It may be that you're trying to pack too much information into tables, or that a tabular format isn't the best way to present some of your data. I would think that using colors and shading and/or patterns like hatching and stippling would be a big help, but you note that you've tried that and it doesn't work. Perhaps disaggregating your tables, possibly using Supplementary Materials, would improve things.	Thank you for further comments and suggestions regarding our manuscript. We are appreciative of the concerns which you have raised. Indeed, we acknowledge that, in having tried to give full hearing to our comprehensive and quite extensive dataset, we ended up trying to fit too much into some of the Tables. Together with the use of notation that is perhaps less familiar to a general audience, this has resulted in a loss of clarity and ease of understanding. We have therefore endeavored to overhaul the Tables, notably, by removing the use of any such superscripts – the prior versions have still been made available in full in a revised version of the Supplementary Materials for any interested readers. As such, the revised Tables (and Figures) can better act in a standalone fashion. We have also reduced as much as possible the use of any abbreviations, moved information about the scales employed for a specific variable from the explanatory note into the Table itself (or its heading), changed titles to be more descriptive of the main results, and employed a similar color scheme in all Tables (i.e., as was previously used only for Table 1). We are confident that all these changes have substantially improved the clarity of the Tables with minimal (if any) reduction in the insights provided.
Take Tables 3-5 for example:  • In Table 3, “a,b,c” seem to signify different levels of support across technologies, in Table 4 “a,b,c” seem to signify different levels of support across cohorts, while in Table 5 “a,b,c” seem to signify different levels of support across policies. • In Table 3, “a,b,c” seem to be relevant only within columns and not across columns (even though the same letter may be used in 	Thanks – as we note above, all superscripts have been removed from the main text, along with the use of such notation being replaced (please see revised Tables 2-4, Table 6) with alternate means to convey findings of statistical significance regarding differences between two (or more) groups.

multiple columns), whereas in Tables 4 and 5 “a,b,c” seem to be relevant only across columns and not within columns (even though in Table 5 the same letter may be used in multiple rows).  • In Table 3, “a,b,c” seem to signify increasing levels of support, but it is not clear if that is the case in Tables 4 and 5. • In Table 3, “1,2” seem to signify different levels of support across cohorts within Table 3, whereas in Table 5 “1,2” seem to signify different levels of support across cohorts in Tables 5a and 5b. 	
The other tables have similar issues of inconsistent signification and orientation and imprecision leading to confusion. I think you need to alter your approach to data presentation in section 3, especially the tables. It may be helpful to bring in an additional coauthor with experience in communicating complex data in understandable ways. At present, I fear that many readers will find these tables and associated results confounding at best, and incomprehensible at worst. Unless these are revamped, I cannot support publication.	All Tables, and indeed our approach to data presentation in Section 3, have been revised in line with the Reviewer’s comments. We would note that our decision to do so, along with our discussion over potential solutions, has been assisted by reaching out to four researchers with extensive expertise (not part of the author team) on such matters to solicit their feedback.
My remaining comments are not dealbreakers, but it would be nice if they were addressed where possible, particularly the final two.	
You respond to my call for discussing greater support for SRM in the Global South than the Global North more extensively by pointing to Table 4 and the related discussion. Fair enough but note that the reason discussion of SRM is not always apparent is because you have chosen to lump it together with CDR as “climate intervention technologies,” which tends to obscure some of the analysis. For example, in practice, discussions about a moratorium or ban focus almost exclusively on SRM, not CDR, yet you discuss it broadly in terms of “technology”—by lumping SRM and CDR together, the potential policy relevance of findings related to proposed bans on SRM are less clear to the reader. Your	Thank you for your comments. We take your point and are fully aware of the broader debate around discussing SRM and CDR together. A few points of clarification seem in order here. First, we employ “climate intervention technologies” in keeping with the frequent use of this term in the literature, for instance in the National Academies of Science report(s) around this topic (NAS 2018, 2021). Our intention here is not to obscure any part of the analysis, but rather to provide a full and fair explication of the results – that is, for both SRM and CDR.

statement that “the findings underscore that it is neither desirable nor illustrative to speak of CDR as a broad category” applies even more strongly to “climate intervention technologies” as a broad category.	Second, while discussions in the policy and expert sphere might (at least at present) focus on SRM, this need not be true for the public at large. The fact that public perceptions remain under-researched, particularly in the Global South, thus advocates for not making presumptions in advance of what issues do or do not matter to them, nor in transposing the views of experts and policymakers onto them. In fact, the support for an international ban or moratorium across the three technology categories is not found to be significantly different – this is despite each participant being randomly assigned to receive information about one of the categories. We cannot therefore agree that the policy relevance of these findings is necessarily diminished. References: National Academies of Sciences, E. (2018). Negative Emissions Technologies and Reliable Sequestration: A Research Agenda. National Academies of Science: Washington, DC. https://doi.org/10.17226/25259 National Academies of Sciences, E. (2021). Reflecting Sunlight: Recommendations for Solar Geoengineering Research and Research Governance. National Academies of Science: Washington, DC. https://doi.org/10.17226/25762
It's still not clear whether Figure 3 is intended to accompany the paragraph immediately preceding it. If it is, that is confusing since the text refers to “composite means for all technologies” but the figure is broken down by individual technology.	Upon further reflection, we agree that this is not the best placement of this Figure. We have moved Figure 3 one paragraph lower in the text, while also adding a specific reference here (which had been missing before). In addition, we have re-drawn the Figure to improve clarity while also breaking down the results in terms of Global North and Global

	South to better correspond to our focus on these differences in the text. We have also revised and shortened the first two paragraphs which focus on the underlying activities related to support (the second of which examines differences between Global North and Global South, and thus provides a bridge to the rest of the sub-section). These revisions are undertaken to reduce any confusion which might exist.
Regarding your response to my comment about BECCS, I think you miss the point—as written, your statement “bioenergy can provide energy for homes and businesses or be stored underground indefinitely” suggests bioenergy can be stored underground indefinitely, but what you mean is carbon. Further, if your statement is revised to read “bioenergy can provide energy for homes and businesses or carbon can be stored underground indefinitely,” this is still not correct as BECCS is envisioned as doing both (with an energy penalty), not one or the other.	We take your point – our previous response focused on the need to avoid giving the sense that bioenergy and CCS could be perfect complements. Having conveyed this to the participants would have risked biasing their responses. Regarding the specifics of the BECCS process, the information text could indeed have used a different phrasing, which might have been clearer. In any case, it is obviously not possible to make changes at this point to the information presented to participants. We would however note that, first, what is ultimately important is whether this phrasing ended up biasing responses – we see no evidence or explanation for why this would be the case. Second, while more explicit details of the underlying process could be provided to participants, this is (i) undeniably true for all approaches and (ii) inherently difficult in light of time and space constraints as well as the need to avoid overloading participants. And, third, though admittedly not BECCS per se, we acknowledge that the somewhat related (and novel) approach of biomass with carbon removal and storage (BiCRS) is becoming more prominent – and where biomass / bioenergy crops are being buried directly.
Reviewer #3: I thought this revision did a good job addressing the concerns raised by	Thank you very much for your kind words and support – we believe that the manuscript

the reviewers, and that this version would be good to publish.	has improved greatly through your comments and attention.
I am going to admit to not being totally won over by the explanatory notes in the tables - they still leave me a bit confused - but will go with it. If it seems to be too wordy / too much info, maybe a simplified version could be presented here with a more detailed version in the SI.	Yes, in response to further comments from another Reviewer, we have undertaken a deep discussion of how best (and simply) for the Tables to convey the main takeaways of the paper – indeed, we even reached out to four researchers with extensive expertise on such matters to solicit their feedback. As a result, we have endeavored to overhaul the Tables, notably, by removing the use of any superscripts – as per your suggestion, prior versions have been made available in full in a revised version of the Supplementary Materials for any interested readers. So as Tables (and Figures) can better act in a standalone fashion, we have also reduced as much as possible the use of any abbreviations, moved information about the scales employed for a specific variable from the explanatory note into the Table itself (or its heading), changed titles to be more descriptive of the main results, and employed a similar color scheme in all Tables (i.e., as was previously used only for Table 1). We are confident that all these changes will substantially improve the clarity of the Tables with minimal (if any) reduction in the insights provided.
I appreciate the new final paragraph, but have to admit that the last sentence kind of opens up a whole new avenue — what does "direct engagement with representative publics in the Global South" entail and how can it be done? It's the kind of statement that makes you think a whole new chapter needs to be written and that more is coming, rather than the kind of statement that neatly wraps a paper. Consider maybe rewording the last clause or adding one more sentence at the end.	Fair point – indeed, this statement was written in light of the further research that we have in mind in this vein. We agree in any case and have added the following sentence at the end: “Future research and endeavors are urgently needed in this vein, especially given the often-dramatic differences in public perceptions and support of climate-intervention technologies in the Global South and Global North.”